# MoRE-Brain: Routed Mixture of Experts for Interpretable and Generalizable Cross-Subject fMRI Visual Decoding

**Yuxiang Wei**
TReNDS Center (GSU, Georgia Tech, Emory)
weiyuxiang@gatech.edu

**Yanteng Zhang**
TReNDS Center (GSU, Georgia Tech, Emory)
yntn32@outlook.com

**Xi Xiao**
University of Alabama at Birmingham
xxiao@uab.edu

**Tianyang Wang**
University of Alabama at Birmingham
toseattle@siu.edu

**Xiao Wang**
Oak Ridge National Laboratory
wangx2@ornl.gov

**Vince D. Calhoun**
TReNDS Center (GSU, Georgia Tech, Emory)
vcalhoun@gatech.edu

## Abstract

Decoding visual experiences from fMRI offers a powerful avenue to understand human perception and develop advanced brain-computer interfaces. However, current progress often prioritizes maximizing reconstruction fidelity while overlooking interpretability, an essential aspect for deriving neuroscientific insight. To address this gap, we propose MoRE-Brain, a neuro-inspired framework designed for high-fidelity, adaptable, and interpretable visual reconstruction. MoRE-Brain uniquely employs a hierarchical Mixture-of-Experts architecture where distinct experts process fMRI signals from functionally related voxel groups, mimicking specialized brain networks. The experts are first trained to encode fMRI into the frozen CLIP space. A finetuned diffusion model then synthesizes images, guided by expert outputs through a novel dual-stage routing mechanism that dynamically weighs expert contributions across the diffusion process. MoRE-Brain offers three main advancements: First, it introduces a novel Mixture-of-Experts architecture grounded in brain network principles for neuro-decoding. Second, it achieves efficient cross-subject generalization by sharing core expert networks while adapting only subject-specific routers. Third, it provides enhanced mechanistic insight, as the explicit routing reveals precisely how different modeled brain regions shape the semantic and spatial attributes of the reconstructed image. Extensive experiments validate MoRE-Brain's high reconstruction fidelity, with bottleneck analyses further demonstrating its effective utilization of fMRI signals, distinguishing genuine neural decoding from over-reliance on generative priors. Consequently, MoRE-Brain marks a substantial advance towards more generalizable and interpretable fMRI-based visual decoding. Codes are publicly available: https://github.com/yuxiangwei0808/MoRE-Brain.

This manuscript has been authored by UT-Battelle, LLC, under contract DE-AC05-00OR22725 with the US Department of Energy (DOE). The US government retains and the publisher, by accepting the article for publication, acknowledges that the US government retains a nonexclusive, paid-up, irrevocable, worldwide license to publish or reproduce the published form of this manuscript, or allow others to do so, for US government purposes. DOE will provide public access to these results of federally sponsored research in accordance with the DOE Public Access Plan (https://www.energy.gov/doe-public-access-plan).

39th Conference on Neural Information Processing Systems (NeurIPS 2025).

# 1 Introduction

Understanding brain visual systems and human perception has been a central interest in the field of neuroscience [1, 2, 3]. The human visual system transforms perceived scenes into rich, meaningful perception via a complex cascade of hierarchical and specialized processing [4]. With the advancement in the diffusion models [5] and representational space like CLIP [6] or visual prompt tuning [7], numerous studies propose to decode visual stimuli from non-invasive brain signals such as functional MRI (fMRI) and EEG, aiming to unravel the complex mechanism under perception and advance the practical brain-computer interface. These methods usually leverage strong priors from existing generative models and train dedicated decoders to decode brain signals to frozen representational space, reconstructing natural scenes with impressive structural accuracy [8, 9, 10, 11, 12, 13].

Despite this progress, a significant gap persists between technical feats and neuroscientific insight. Current leading methods often treat fMRI signals as relatively monolithic inputs, lacking architectural designs that reflect established principles of neural processing. For instance, several works manually segment the brain into broad regions like the higher visual and lower visual cortex, then decode semantic and low-level spatial information from them separately [14, 15, 16, 17]. However, this simplification overlooks the intricate nature of the visual pathway, which involves multiple hierarchical processing stages across numerous specialized cortical areas, each handling different facets of visual information [18, 19]. Furthermore, leading research commonly encounters two critical bottlenecks: cross-subject generalization and limited mechanistic interpretability. Due to inter-individual brain variability, models often require subject-specific training. While approaches like mapping subject data to a shared space [8, 20, 21] or utilizing subject-specific tokens [11] exist, they typically necessitate extra training and are difficult to generalize to new subjects. More fundamentally, the predominant focus on maximizing decoding performance often obscures how different neural computations contribute to the final reconstruction. This lack of transparency hinders our ability to validate these models against neuroscientific knowledge or use them to deepen our understanding of the visual system itself.

To bridge the gap, we introduce MoRE-Brain (Mixture-of-Experts Routed Brain decoder), a novel visual decoding framework that is explicitly designed to mirror the hierarchical and specialized nature of the human visual pathway [22] to overcome key limitations in generalization and interpretability inherent in less biologically constrained approaches. MoRE-Brain builds upon a hierarchical Mixture-of-Experts (MoE) architecture to first map fMRI activity patterns to image and text priors. Instead of relying on a single, monolithic network, MoRE-Brain employs multiple layers of "expert" subnetworks. Within each layer, trainable "router" networks learn to assign specific brain regions (voxel groups) to the experts best suited to process their signals. This allows each expert to specialize in decoding information from distinct neural populations, analogous to functional specialization in the cortex. Addressing the challenge of cross-subject generalization, we draw from findings suggesting that significant inter-individual variation arises from spatial differences in functional network topography rather than fundamental computational differences [23, 24]. Based on this, we hypothesize that the core visual decoding computations (the experts) can be shared across subjects, while subject-specific adjustments are primarily needed for the routers that map individual brain topographies to these experts. Our experiments validate this, demonstrating that by freezing expert weights and fine-tuning only the routers, MoRE-Brain achieves strong cross-subject performance with minimal subject-specific data, offering a more scalable approach than retraining entire models.

Furthermore, effectively utilizing the disentangled information captured by these specialized experts within the iterative image generation process of diffusion models presents another challenge. This mechanism operates dynamically across both time and space: (1) A **Time Router** selects which hierarchical *level* of expert embeddings to prioritize at different diffusion timesteps. This potentially reflects the coarse-to-fine dynamics of visual processing, where global scene layout might be established before finer details emerge [25]. (2) A **Space Router** then dynamically modulates the spatial influence of the outputs from individual experts *within the selected level*. This mimics how the brain might integrate features processed by different specialized areas (e.g., form, color, motion) into a spatially coherent percept [26]. Critically, the dual-routing system not only guides image generation but also provides a mechanistic lens into the reconstructing process. By observing which experts and levels are routed at different times and for images with different semantic concepts, we gain insight into how distinct, modeled neural sources dynamically contribute to the final reconstruction.

Our main contributions are summarized as follows:

- We propose MoRE-Brain, an fMRI decoding framework featuring a hierarchical MoE architecture explicitly inspired by the principles of functional specialization and hierarchy in the visual brain.
- MoRE-Brain addresses cross-subject generalization by leveraging the MoE structure, sharing expert networks while adapting subject-specific routers, enabling efficient tuning for new individuals.
- We introduce a novel dynamic Time and Space routing mechanism to effectively integrate information from specialized experts as conditional prompts for diffusion models, reflecting potential temporal and spatial integration processes in vision.
- This routing mechanism enhances interpretability, offering insights into how information from different modeled brain networks dynamically shapes the semantics of reconstructed images throughout the generative process.

## 2 Methodology

### 2.1 Preliminaries: Latent Diffusion Models and Conditioning

Our reconstruction module leverages latent diffusion models (LDMs), specifically Stable Diffusion XL (SDXL) [5], which are generative models operating in a compressed latent space [27, 28]. LDMs learn to reverse a diffusion process that gradually adds Gaussian noise to latent representations ($z_0$) over discrete timesteps $t = 1, \ldots, T$. The forward process is defined as $z_t = \sqrt{\bar{\alpha}_t} z_0 + \sqrt{1 - \bar{\alpha}_t} \epsilon$, where $\bar{\alpha}_t$ is a noise schedule and $\epsilon \sim \mathcal{N}(0, \mathbf{I})$. The core of the LDM is a denoising network $\epsilon_\theta$ (typically a U-Net [29]), trained to predict the noise $\epsilon$ added at timestep $t$, given the noisy latent $z_t$ and conditioning information $c$:

$$\mathcal{L}_{LDM} = \mathbb{E}_{z_0, \epsilon, c, t} \left[ ||\epsilon - \epsilon_\theta(z_t, t, c)||_2^2 \right] \tag{1}$$

Conditioning information $c$ is typically derived from various modalities. For text-to-image synthesis, $c$ often consists of embeddings $\tau_{text}(c_{text})$ from a pre-trained text encoder (e.g., CLIP [6]). Image conditioning can be achieved using IP-Adapter [30], where features $\tau_{img}(c_{img})$ are extracted from a reference image via a pre-trained image encoder (e.g., CLIP image encoder) and mapped through an adapter network. Both $\tau_{text}(c_{text})$ and $\tau_{img}(c_{img})$ are integrated into $\epsilon_\theta$ usually via cross-attention mechanisms. In MoRE-Brain, we adapt this conditioning mechanism to use fMRI-derived embeddings as image and text priors.

### 2.2 MoRE-Brain Framework Overview

MoRE-Brain employs a two-stage process, illustrated in Figure 1, designed to decode visual information from fMRI while incorporating principles of neural specialization and hierarchy.

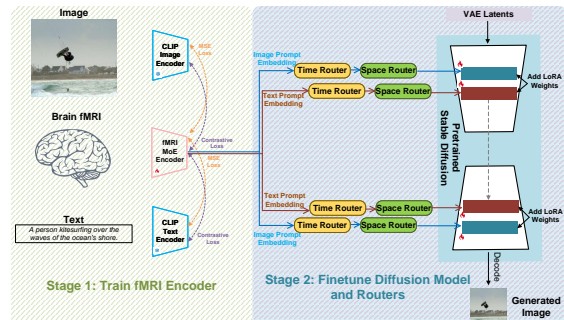

- **Hierarchical MoE for fMRI-to-Embedding Decoding:** A hierarchical Mixture-of-Experts (MoE) maps input fMRI signals ($\mathcal{F}$) to sets of embedding aligned with the frozen CLIP space. This stage learns to disentangle fMRI representations based on functionally specialized voxel groups.

Figure 1: Overview of MoRE-Brain: fMRI is encoded into CLIP space by the hierarchical MoE (left), guiding image generation via dynamic Time/Space Routers (right).

- **Dynamically Conditioned Image Generation:** We introduce a novel Time and Space routing mechanism to dynamically select and integrate the multi-level expert embeddings produced in Stage 1 to guide the diffusion model's (SDXL's) denoising process at each timestep $t$. During this stage, the MoE fMRI encoder (trained in Stage 1) is frozen, while the SDXL U-Net (via LoRA [31]) and the Time and Space routers are fine-tuned.

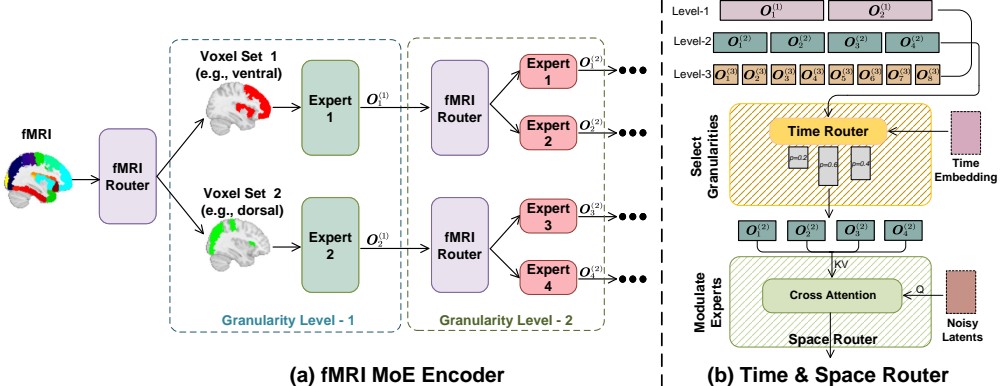

Figure 2: Routing mechanism of MoRE-Brain. (a) Hierarchical fMRI MoE encoder processes voxels using routed, specialized experts across levels. (b) Time and space router that adaptively selects and modulates expert outputs.

## 2.3 Hierarchical MoE for fMRI Encoding

To model functional specialization and hierarchy akin to the brain's visual system, we design a hierarchical MoE architecture (Figure. 2a) to transform fMRI signals into sets of CLIP-aligned embeddings. Given the input fMRI voxels $\mathcal{F} \in \mathbb{R}^v$, where $v$ is the number of voxels, instead of pre-defining brain parcels, we learn learn data-driven voxel assignments. As in Figure. 2(a), for each level of the hierarchy, a router network determines the affinity of each voxel for each expert at that level. Concretely, at level $l$, given input features $X^{(l)}$ (where $X^{(0)} = \mathcal{F}$), the router computes voxel-expert affinity scores $A^{(l)} \in \mathbb{R}^{v \times e_l}$ using a learnable weight matrix $W_r^{(l)}$:

$$A^{(l)} = W_r^{(l)} X^{(l)} \tag{2}$$

where $e_l$ is the number of experts at level $l$. We then apply softmax over the expert dimension to get probabilities $P^{(l)} = \text{softmax}(A^{(l)})$. To assign voxels to experts, we employ Top-K selection for each expert. For the $j$-th expert at level $l$, we select the indices $I_j^{(l)}$ of the $k$ voxels with the highest probability scores in the $j$-th column of $P^{(l)}$:

$$(S_j^{(l)}, I_j^{(l)}) = \text{TopK}(P_{:,j}^{(l)}, k) \tag{3}$$

where $k = \lfloor \frac{v}{e_l} \times c_f \rfloor$ is the number of voxels assigned per expert, controlled by a capacity factor $c_f$. We set $c_f = 1$ to enable non-overlapping selection over all voxels. Each expert $E_j^{(l)}$ is implemented as a simple MLP, processing only the features corresponding to its assigned voxels $I_j^{(l)}$.

For progressively finer specialization, outputs $O_j^{(l)}$ from experts at level $l$ serve as input $X^{(l+1)}$ to the routers at the next level $(l+1)$. We start with $e_0 = 2$ experts at the first level, and each expert's output path is routed to 2 new experts at the subsequent level. We use a total of $L = 4$ levels, resulting in $2^4 = 16$ experts at the final level. This number was empirically chosen to approximate the number of distinct functional ROIs identified in recent visual cortex atlases [32]. The expert outputs are subsequently aligned with frozen CLIP embeddings, more details in Appendix A.

## 2.4 Dynamic Conditioning via Time and Space Routing

Generating images via diffusion is an iterative process. We leverage the hierarchical and specialized nature of the MoE embeddings dynamically, which allows the model to potentially focus on different levels of abstraction (semantic vs. detail) processed by different specialized networks at different stages of generation, mirroring coarse-to-fine processing theories [33]. We introduce a time and space routing mechanism (Figure 2b) to integrate MoE expert embeddings ($E$) into SDXL U-Net ($\epsilon_\theta$).

The time router $\mathcal{R}_{\mathcal{T}}$ determines the relevance of expert embeddings from different hierarchical *levels* based on the current diffusion timestep $t$. We define learnable embeddings $\Phi = [\phi_1, ..., \phi_L]$

representing each of the $L$ MoE levels, and compute the relevance of each level $l$ for each timestep $t$ via an attention-like mechanism:

$$P_T = \text{softmax}\left(\frac{(W_Q t_c) \cdot (W_K \phi)^T}{\sqrt{d_k}}\right) \tag{4}$$

where $W_Q$ and $W_K$ are learnable matrices, $t_c$ is the continuous representation of $t$, and $P_T \in \mathbb{R}^L$ contains weights indicating the importance of each level at timestep $t$. To explicitly encourage coarse-to-fine processing, the $P_T$ is regularized by a guiding distribution $\bar{P}_T$ via KL divergence. For total $T$ steps, we define $\bar{P}_T$ using a Gaussian centered at $\mu_t$:

$$\bar{P}_{T,l} = \frac{\exp\left(-\frac{(l-\mu_t)^2}{2\sigma^2}\right)}{\sum_{k=1}^{L} \exp\left(-\frac{(k-\mu_t)^2}{2\sigma^2}\right)} \quad \text{where} \quad \mu_t = L \cdot \frac{t}{T} \tag{5}$$

Here, $\sigma = 1$ controls the sharpness of the focus on the target level $\mu_t$. Based on $P_T$, we explore two selection methods: (1) *soft selection* by applying $P_T$ to weigh the expert embedding $O^{(l)}$ at the corresponding level and (2) *hard selection* by choosing level $l^*$ with the highest weight ($l^* = \text{argmax}_l P_{T,l}$). We further provide a fixed schedule by predefining timestep intervals according to each level, as detailed in Appendix A.4.

After selecting the relevant expert embeddings ($E_{sel}$ based on $\mathcal{R}_\mathcal{T}$'s output), the space router $\mathcal{R}_\mathcal{S}$ modulates their influence spatially based on the current state of the noisy latents $z_t$. This mimics integrating specialized features (e.g., shape, texture) into a coherent spatial layout. We use a cross-attention mechanism where $z_t$ acts as the query and the selected expert embeddings $E_{sel}$ act as keys and values:

$$C = \text{softmax}\left(\frac{\hat{W}_Q z_t \cdot \hat{W}_K E_{sel}}{\sqrt{d_k}}\right) \cdot \hat{W}_V E_{sel} \tag{6}$$

The resulting conditioning $C$ serves as the dynamic, fMRI-derived condition for the SDXL U-Net's cross-attention layers at timestep $t$.

## 3 Experiments

### 3.1 Settings and Implementations

We evaluate MoRE-Brain using the Natural Scenes Dataset (NSD) [34], a large-scale fMRI dataset comprising brain responses from 8 subjects viewing over 30,000 distinct natural images. We adhere

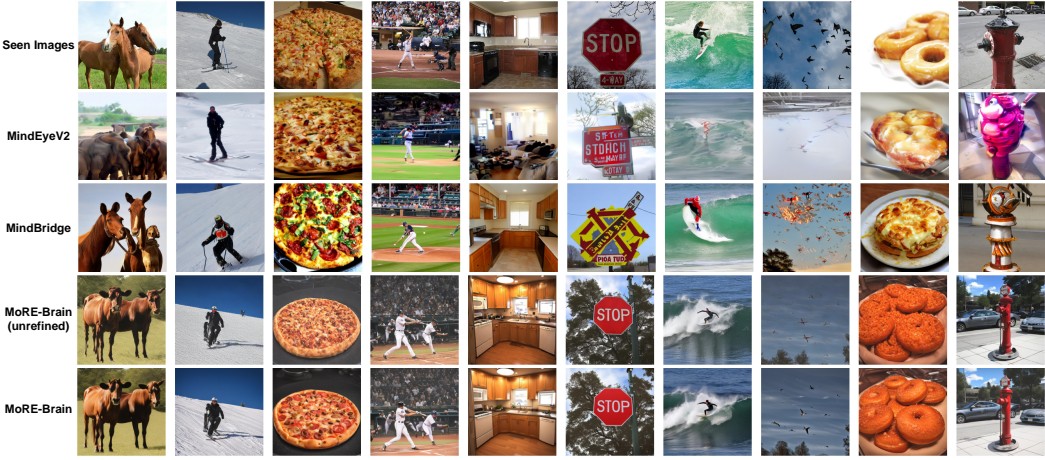

Figure 3: Reconstructions from different methods on the subject 1. More-Brain can reconstruct visual stimuli faithfully compared to baselines. An optional refinement step (see Appendix A.1.3) is employed to enhance the image.

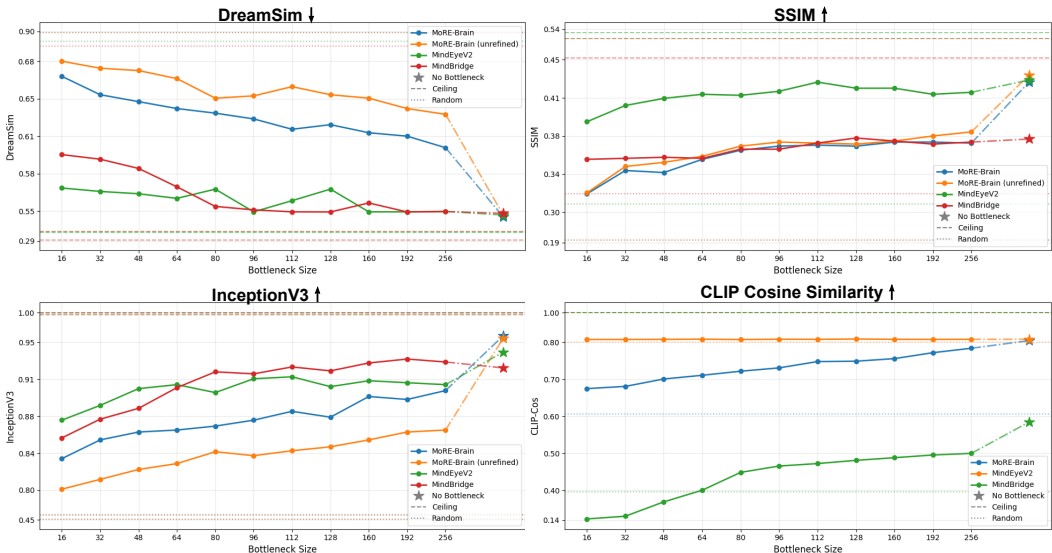

Figure 4: Quantitative performance across varying bottleneck sizes. MoRE-Brain shows a consistent and notable performance drop across most metrics as the bottleneck size decreases. Moreover, its performance witnesses the most significant drop after adding a bottleneck, indicating that the bottleneck is particularly detrimental to the learned rich information and the model heavily relies on the fMRI data. However, MindEye2 and MindBridge exhibit less degradation, particularly MindBridge on SSIM and InceptionV3, suggesting a greater influence of learned priors. Notably, MindEye2 maintains high CLIP-Cos even at extreme bottlenecks, while others show expected decline with information restriction.

to the standardized training and testing splits proposed in prior work [8]. Further details on NSD and our fMRI data preprocessing pipeline are available in Appendix B.

We compare MoRE-Brain with two methods: the recent state-of-the-art MindEye2 [8] and Mind-Bridge [35]. To assess the extent to which models leverage fMRI information versus relying on generative priors, we adopt the bottleneck analysis proposed by [10], evaluating performance across varying bottleneck sizes imposed on the fMRI-derived features. We include high-level metrics like DreamSim [36] and InceptionV3 [8, 37], and a low-level metric SSIM. We also measure the cosine similarity between decoded embeddings and the ground truth CLIP embedding as in [10], thereby isolating the decoder's performance from the generative model. For each method, we report a "reconstruction ceiling" (using ground truth image latents instead of fMRI-predicted latents) and a "random baseline" (using random fMRI as input). More details on implementation in Appendix A.

## 3.2 fMRI-to-Image Reconstruction

First, we evaluate the overall capabilities of fMRI-to-image reconstruction. Figure 3 presents qualitative examples of reconstructions for Subject 1, where MoRE-Brain faithfully recovers visual stimuli. Quantitatively, Figure 4 shows average performance across all subjects. MoRE-Brain's performance is competitive with state-of-the-art methods. We also show reconstructions from a single level of expert embeddings, see Appendix E.

The bottleneck analysis in Figure 4 provides critical insights. As highlighted in the caption, MoRE-Brain's reconstruction quality (SSIM, DreamSim, InceptionV3) is notably more sensitive to the amount of information allowed through the bottleneck compared to MindEye2 and MindBridge. This suggests MoRE-Brain effectively utilizes the richness of the fMRI signal. On the other hand, the baselines show relative robustness to narrow bottlenecks in terms of pixel-space metrics, especially MindBridge. Surprisingly, MindEye2 exhibits a remarkably high random baseline CLIP-Cos of approximately 0.8, and crucially, its CLIP-Cos performance remains largely consistent and high across *all* tested bottleneck sizes, showing minimal degradation even with severe information restriction. We hypothesize this is due to MindEye2 learns strong biases towards semantic plausibility, as defined by CLIP, and overshadows the contribution of fMRI signals. Although it generates embeddings that

Table 2: Ablation study of the text/image conditioning and the time and space routers based on unrefined MoRE-Brain. $\mathcal{T}$: Text conditioning only. $\mathcal{I}$: Image conditioning only. (TS): Both Time and Space routers. (S): Only Space router. (T, sched): Time router with fixed schedule. (T, hard): Time router with learned hard selection. (T, soft): Time router with learned soft selection.

| | $\mathcal{T}$ | $\mathcal{T}$ (TS) | $\mathcal{I}$ | $\mathcal{I}$ (TS) | $\mathcal{T}/\mathcal{I}$ | $\mathcal{T}/\mathcal{I}$ (S) | $\mathcal{T}/\mathcal{I}$ (T, sched) | $\mathcal{T}/\mathcal{I}$ (T, hard) | $\mathcal{T}/\mathcal{I}$ (T, soft) | $\mathcal{T}/\mathcal{I}$ (TS) |
|---|---|---|---|---|---|---|---|---|---|---|
| SSIM ↑ | 0.410 | 0.402 | 0.422 | 0.382 | 0.407 | 0.375 | 0.403 | 0.397 | 0.402 | **0.415** |
| Alex (2) ↑ | 0.748 | 0.754 | 0.742 | **0.802** | 0.760 | 0.762 | 0.765 | 0.650 | 0.764 | 0.792 |
| Incep ↑ | 0.926 | 0.948 | 0.893 | 0.957 | 0.890 | 0.943 | 0.940 | 0.867 | 0.941 | **0.962** |
| DreamSim ↓ | 0.560 | 0.542 | 0.605 | 0.528 | 0.585 | 0.546 | 0.533 | 0.626 | 0.543 | **0.507** |

make semantic sense to CLIP for any input, the structural information that depends on the fMRI is removed, leading to the relatively low random performance as measured by other metrics.

### 3.3 Cross-Subject Generalization with Limited Data

MoRE-Brain's separation of shared experts from subject-specific routers allows for efficient generalization with limited data. After training the model on subject 1, we freeze the model's experts and only finetune the routers to generalize to other subjects. This massively reduces the parameters during finetuning. We compare the total parameters of MoRE-Brain and the two baselines, as in Table 1. Note that for MindBridge, we applied its 'reset-tuning' strategy.

To further demonstrate MoRE-Brain's generalizing performance with limited data, we first train the full model from scratch on subject 1, then freeze the weights of all experts and finetune only the fMRI routers (Eq. 2) with 2.5% (1 session), 10% (4 sessions), 25% (10 sessions), 50% (20 sessions), and all new subjects' data (40 sessions). See results in Appendix C.

Table 1: Compare the total number of parameters and trainable parameters during finetuning.

| | Total Params | Trainable |
|---|---|---|
| MindEve2 | 729.3M | 100% |
| MindBridge | 552.9M | 98.48% |
| MoRE-Brain | 293.4M | 44.84% |

### 3.4 Ablations

To understand the contributions of MoRE-Brain's key components, we conduct ablation studies and the results are shown in Table 2. Since the cosine similarity mentioned before does not apply here, we include another low-level metric Alex (2).

Comparing text-only ($\mathcal{T}$) versus image-only ($\mathcal{I}$) conditioning, we find that image conditioning generally yields superior SSIM, reflecting better structural detail. For the time router, the fixed schedule (see Appendix A.4) enhances the reconstruction compared to no router, while being worse than the learned soft selection. Overall, the results affirm the importance of both the dual conditioning streams and the dynamic routing mechanisms.

### 3.5 Interpretability: Linking Model to Brain Function

A core motivation for MoRE-Brain is to enhance neuroscientific insight. We investigate model interpretability at two levels: overall brain contributions and expert-specific functional specialization. We show MoRE-Brain learns neurophysiologically plausible mappings (Figure 5, 7), with its experts learn hierarchical spatial specialization (Figure 6) and increasingly specific semantic preferences (Figure 8), mimicking aspects of brain

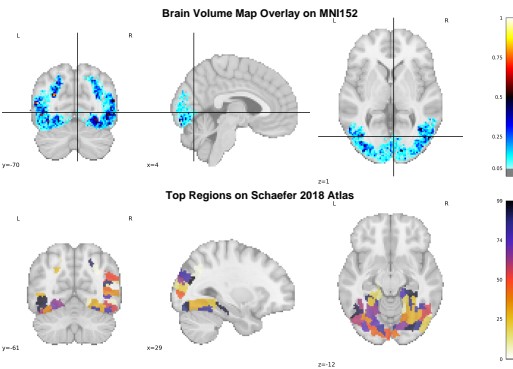

Figure 5: Interpretability of overall model contributions via ICA. Brain regions consistently contributing to reconstructions across 8 subjects are identified as Independent Components (ICs). ICs highlight engagement of known visual processing areas (e.g., visual central/periphereal) and higher-order association areas (e.g., dorsal attention network). This demonstrates the MoRE-Brain learns neurophysiologically plausible mappings. See Appendix G for complete visualizations of all ICs and ROI labels.

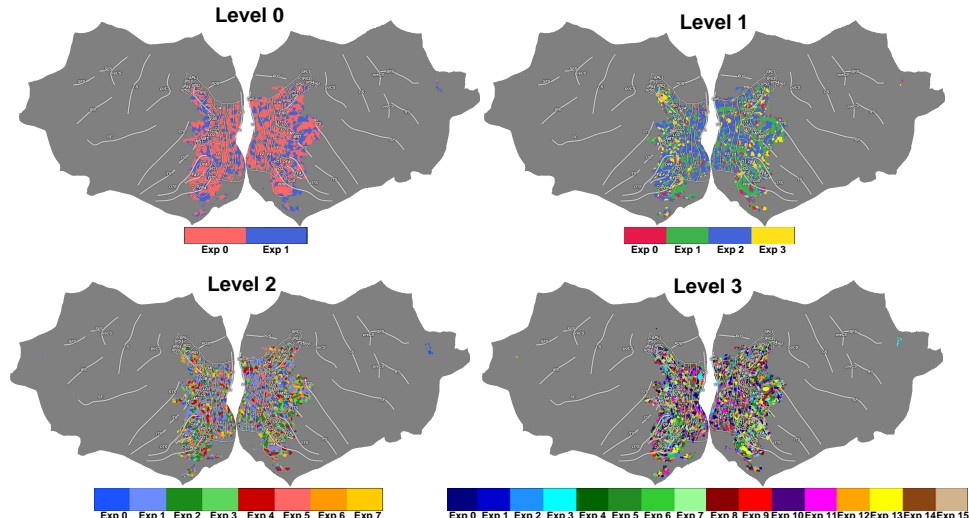

Figure 6: Visualization of voxel selectivity for individual experts within MoRE-Brain's hierarchical MoE fMRI encoder (Subject 1, attributions projected to fsaverage surface). Each panel shows the contributing voxels for a specific expert $E_j^{(l)}$. 1) *hierarchical specialization:* Experts at early levels (e.g., level 0, $E_0^{(0)}, E_1^{(0)}$) tend to draw from broader visual regions. Experts at higher levels (e.g., level 2, level 3) exhibit finer-grained spatial specialization, focusing on more distinct sub-regions of the visual cortex and associated areas. 2) *intra-level differentiation:* Within a single level, different experts (e.g., $E_0^{(2)}$ vs. $E_1^{(2)}$) show preferences for different voxel populations, supporting the MoE's role in disentangling neural signals. Voxel assignments appear relatively balanced. See Appendix J for detailed volume-based visualizations and ROI labels.

organization. More analyses on the routing mechanism and the expert's functional specialization can be found in Appendix F-I.

### 3.5.1 Overall Brain Contributions to Reconstruction

To identify brain regions crucial for reconstruction across subjects, we compute voxel attributions using GradientSHAP [38] for the entire MoRE-Brain. The attribution maps (1000 per subject, 8000 total) are then decomposed using Independent Component Analysis (ICA) [39] into 32 spatial ICs, representing distinct co-varying patterns of functional activity contributing to the decoding.

Furthermore, by correlating IC activations (derived from the ICA mixing matrix) with the semantic content of the input stimuli (12 COCO supercategories [40]), we find that specific ICs exhibit significant preferences for object categories (Figure 7). For example, ICs strongly correlated with "appliance" consistently involve parieto-occipital regions. (Details on IC labeling with Schaefer atlas [41] in Appendix G).

### 3.5.2 Expert-Specific Functional Specialization

MoRE-Brain's hierarchical MoE architecture is designed to allow experts to specialize. We

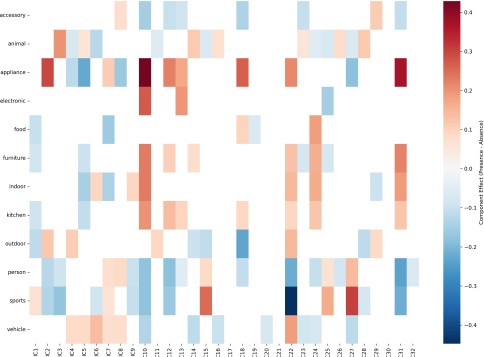

Figure 7: Semantic correlates of ICs. Heatmap shows significant (FDR corrected t-test) correlations between IC activations (from ICA mixing matrix) and the 12 COCO supercategories in stimuli. Different ICs show preferential responses to semantic categories (e.g., ICs 2, 10, 18, 31 correlating with "appliance" map to visual and parieto-occipital areas). This links distributed brain activity patterns captured by ICs to high-level semantic features. More in Appendix I.

investigate this by examining the voxel selections and semantic correlations of individual experts. Figure 6 illustrates that experts progress from integrating broader visual regions at early MoE levels to focusing on more distinct voxel populations at higher levels, with clear differentiation also observed between experts within the same level.

This functional specialization extends to semantic processing. Figure 8 shows the correlation between expert contributions and the 12 COCO supercategories. While early-level experts show diffuse category preferences, higher-level experts increasingly specialize towards particular semantic categories. For instance, expert $E_2^{(2)}$ (level 2) shows a notable preference for "outdoor" scenes, primarily processing signals from the dorsal attention network and parts of the default mode network. Some experts remain more "universal." These findings support the hypothesis that MoRE-Brain learns functional specialization and hierarchical processing analogous to the visual brain. (Further visualizations and analyses are provided in Appendix J).

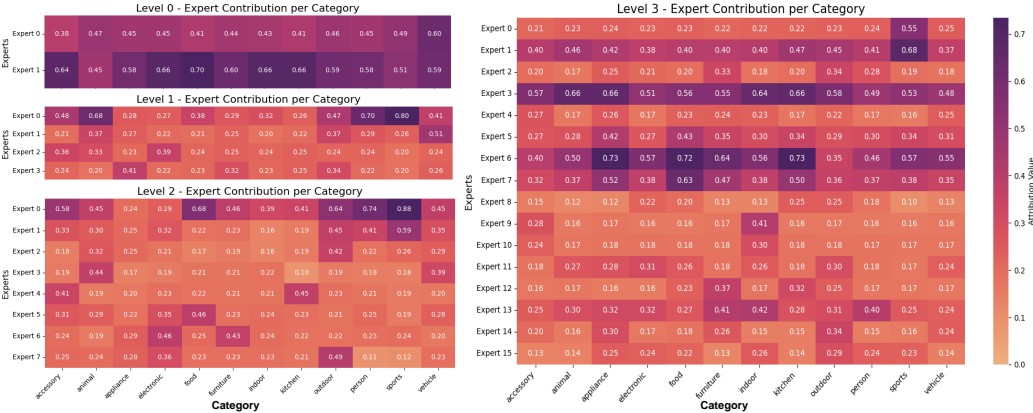

Figure 8: Semantic selectivity of individual experts across MoE levels. Heatmap shows the average attribution (contribution score) of each expert $E_j^{(l)}$ to reconstructions of images containing the 12 COCO supercategories. 1) *emergent specialization:* Experts at the initial level (level 0) show broad, less differentiated responses across categories. 2) *higher-level selectivity:* Experts at higher levels (e.g., level 2, level 3) develop stronger preferences for specific semantic categories (e.g., $E_2^{(2)}$ for "outdoor," certain level 3 experts for "appliance" or "food"). Some experts (e.g., $E_0^{(1)}$, some level 3 experts) remain more "universal," contributing to multiple categories, possibly encoding foundational visual features. This suggests an emergent functional hierarchy within the MoE.

## 4 Conclusion

Decoding visual experiences from fMRI remains challenged by cross-subject variability and the "black box" nature of many models. In this work, we presented MoRE-Brain, a framework that directly tackles these issues through neuro-inspired design. MoRE-Brain's hierarchical MoE fMRI encoder explicitly models the brain's specialized processing networks. This architecture, combined with a dynamic dual-stage routing system guiding a powerful diffusion model, achieves high-fidelity reconstruction. More importantly, the MoE structure serves a dual purpose: it facilitates efficient cross-subject generalization by isolating subject-specific routing from shared expert computations, and it unlocks unprecedented mechanistic interpretability. Our bottleneck analyses validate MoRE-Brain's reliance on genuine neural information, while detailed interpretability studies reveal emergent functional specialization within the experts and highlight the dynamic contributions of different modeled brain regions to semantic and spatial aspects of the reconstruction. By bridging the gap between decoding performance and neuroscientific understanding, MoRE-Brain represents a significant step towards developing fMRI decoding systems that are both highly adaptable and interpretable, paving the way for their use as effective tools in cognitive neuroscience.

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

# A More Details on MoRE-Brain

## A.1 Training and Inference

MoRE-Brain's training is conducted in two distinct stages:

### A.1.1 Stage 1: Training the Hierarchical MoE fMRI Encoder

In the first stage, we train the hierarchical Mixture-of-Experts (MoE) fMRI encoder (described in Section 2.3) to map fMRI signals to the embedding spaces of the frozen CLIP image (ViT-bigG/14) and text encoders (from LAION-2B [42, 6]). The training objective for this stage adapts losses from [8] and incorporates a load balancing loss for the MoE routers:

$$\mathcal{L}_{\text{Stage1}} = \alpha_1 \mathcal{L}_{\text{MSE}}(\text{Pred}_{\text{CLIP}}, \text{GT}_{\text{CLIP}}) + \alpha_2 \mathcal{L}_{\text{Contrastive}}(\text{Pred}_{\text{CLIP}}, \text{GT}_{\text{CLIP}}) + \alpha_3 \sum_{l=0}^{L-1} \mathcal{L}_{\text{LB}}^{(l)} \qquad (7)$$

where:

- $\text{GT}_{\text{CLIP}}$ represents the ground truth CLIP image and text embeddings corresponding to the viewed stimulus.

- $\text{Pred}_{\text{CLIP}}$ is the set of predicted CLIP embeddings generated by the MoE fMRI encoder. To obtain a single prediction for the loss calculation in this stage, we aggregate the CLIP-aligned embeddings produced by the experts. Specifically, for each level $l$, the embeddings $\{(c_{l,j}^{img}, c_{l,j}^{text})\}_{j=1}^{e_l}$ from all its $e_l$ experts are averaged. Then, these level-wise average embeddings are themselves averaged across all $L$ levels to produce the final $\text{Pred}_{\text{CLIP}}$:

$$\text{Pred}_{\text{CLIP}}^{\text{type}} = \frac{1}{L} \sum_{l=0}^{L-1} \left( \frac{1}{e_l} \sum_{j=1}^{e_l} c_{l,j}^{\text{type}} \right) \qquad (8)$$

where type is either 'img' or 'text'. This aggregation is used for computing $\mathcal{L}_{\text{MSE}}$ and $\mathcal{L}_{\text{Contrastive}}$.

- $\mathcal{L}_{\text{MSE}}$ is the Mean Squared Error loss.

- $\mathcal{L}_{\text{Contrastive}}$ is a contrastive loss (i.e., InfoNCE) incorporating MixCo augmentation [43] and soft target labels (derived from the dot product similarity matrix of ground truth CLIP embeddings within a batch), similar to SoftCLIP in [8]. This loss is applied separately for image and text embeddings.

Table 3: Using text embeddings projected from image embeddings to condition the diffusion model.

| | SSIM | Alex (2) | Incep | DreamSim |
|---|---|---|---|---|
| baseline | 0.415 | 0.792 | 0.962 | 0.507 |
| project-based text conditioning | 0.384 | 0.773 | 0.941 | 0.542 |

- $\mathcal{L}_{\text{LB}}^{(l)}$ is a load balancing loss for the router at level $l$, encouraging a more even distribution of voxel assignments across its $e_l$ experts. It is defined based on the fraction of voxels $f_j^{(l)}$ assigned to each expert $j$ at level $l$:

$$L_{LB}^{(l)} = \sum_{j=1}^{e_l} \left( f_j^{(l)} - \frac{1}{e_l} \right)^2, \quad \text{where } f_j^{(l)} = \frac{\text{count of voxels routed to expert } j \text{ at level } l}{\text{total number of voxels } v} \tag{9}$$

We empirically set $\alpha_1 = 1$, $\alpha_2 = 0.33$, and $\alpha_3 = 0.1$.

### A.1.2 Stage 2: Fine-tuning SDXL with Time and Space Routers

In the second stage, the pre-trained MoE fMRI encoder from Stage 1 is frozen. We then train the Time and Space Routers (Section 2.4) and fine-tune the SDXL diffusion model. Low-Rank Adaptation (LoRA) [31] with a rank of 16 is applied to the cross-attention layers of the SDXL U-Net to enable efficient fine-tuning. The training objective for this stage is the standard LDM denoising loss (as in Eq. 1):

$$\mathcal{L}_{\text{Stage2}} = \mathbb{E}_{z_0, \epsilon, \mathcal{F}, t} \left[ ||\epsilon - \epsilon_\theta(z_t, t, C_{fMRI})||_2^2 \right] \tag{10}$$

where $C_{fMRI}$ is the conditioning derived from fMRI signals $\mathcal{F}$ via the frozen MoE encoder and the trainable Time and Space Routers, as described in Section 2.4. At each training iteration, a random diffusion timestep $t$ is sampled.

### A.1.3 Inference

During inference, the frozen MoE fMRI encoder processes the input fMRI data to produce multi-level sets of CLIP embeddings $C^{(l)} = \{(c_{l,j}^{img}, c_{l,j}^{text})\}_{j=1}^{e_l}$ for each level $l \in \{0, \dots, L-1\}$. The Time Router selects a level $l^*$, and the Space Router then uses the embeddings $C^{(l^*)}$ from this chosen level to generate the dynamic conditioning $C_{fMRI}$ for the fine-tuned SDXL model at each denoising step. We use a classifier-free guidance scale of 15 and a maximum of 30 denoising steps for image generation.

After generating the image, we include an optional refinement step as in [8]. We use the refinement model of SDXL, with the same text guidance as the first generation, to refine the model. We empirically find this slightly enhances the reconstruction quality (see Figure 3, 4).

### A.2 fMRI-to-CLIP Text Embedding

The text captions associated with stimuli (i.e. COCO captions) vary in length. The full CLIP text embedding, typically $\text{CLIP}_{\text{Text}} \in \mathbb{R}^{N_{tokens} \times D_{CLIP}}$ (e.g., $N_{tokens} = 77$, $D_{CLIP} = 1280$), contains information distributed in the first few token positions. However, [44] suggests that for conditional diffusion models, the embedding of the end-of-sequence (EOS) token often carries a significant portion of the overall semantic information. Based on this, for our fMRI-to-text decoding, we primarily focus on predicting the EOS embedding from the fMRI signal: $\text{Pred}_{\text{EOS}} \in \mathbb{R}^{1 \times D_{CLIP}}$. During Stage 1 training, the $\text{Pred}_{\text{CLIP}}^{\text{text}}$ (Eq. 8) is trained to match the ground truth EOS embedding of the corresponding caption.

To use the predicted EOS embeddings for conditioning, we expand the single predicted EOS vector $\text{Pred}_{\text{EOS}}$ to fill the first $N_{tokens} - 1$ positions (i.e., 76 positions) of the text conditioning tensor. The final token position is then filled with a fixed start-of-sequence (SOS) token obtained from the frozen CLIP model. This approach empirically yields strong decoding performance while reducing the complexity of the fMRI-to-text mapping task.

### A.3 Alternative: Projecting Image Embeddings to Text Space

Inspired by [44], which demonstrated that CLIP image embeddings can be effectively projected into the CLIP text embedding space via a simple linear transformation, we also explore an alternative strategy to generate text conditioning without training another decoder. In this setup, we directly use the closed-form linear transformation matrix from [44] to generate $\text{Pred}_{\text{EOS}}$. Our experiments in Table 3 indicate that this projection method achieves comparable performance to direct fMRI-to-text decoding.

## A.4 Fixed Time Routing Schedule

The human visual system is understood to process scenes in a coarse-to-fine manner over time. Inspired by this, one of the strategies for our Time Router (Section 2.4) is a fixed schedule that deterministically selects an expert level $l^*$ based on the current diffusion timestep $t$.

Given $L$ hierarchical levels in the MoE fMRI encoder ($l \in \{0, 1, \ldots, L-1\}$) and a total of $T$ diffusion timesteps ($t \in \{0, 1, \ldots, T-1\}$), the fixed routing schedule selects the level $l^*$ as follows:

$$l^* = \min\left(\left\lfloor \frac{L \cdot t}{T} \right\rfloor, L - 1\right) \tag{11}$$

This schedule proportionally maps the normalized current timestep ($t/T$) to the available MoE levels. Earlier timesteps $t$ (closer to 0) will select lower-indexed levels (e.g., level 0), which are hypothesized to capture coarser, more global information. Later timesteps $t$ (closer to $T-1$) will select higher-indexed levels (e.g., level $L-1$), hypothesized to capture finer details. The $\min(\cdot, L-1)$ ensures the index stays within the valid range $[0, L-1]$. The fixed schedule serves as a simple baseline to validate our intuition over the coarse-to-fine selection strategy, as in Table 2.

## B  NSD Dataset and Preprocessing

The fMRI signals are beta weights estimated from each session by GLMSingle [45]. We use the preprocessed fMRI voxels in 1.8mm volume space, masked by the "nsdgeneral" ROI mask provided by the NSD dataset. fMRI from subjects 1, 2, 5, and 8 contains 40 sessions, while the other subjects have about 30 sessions. To ensure consistent training data volume across subjects for certain analyses and to simplify batching, we balanced the training sample counts. For subjects with fewer samples, a subset of their existing samples was randomly selected and repeated to match the count of the subject with the maximum number of samples. This balancing was applied only where strictly necessary for model training architecture, while test performance was always evaluated on unique trials. The test set consists of images from NSD's "shared1000" subset, comprising 1000 images viewed by all 8 subjects, ensuring comparability. Following established practices [8], we applied z-score normalization to the fMRI voxel activities. This normalization was performed separately for the training and test sets within each subject to prevent data leakage.

## C  MoRE-Brain's Generalization Ability

We first present the results of finetuning MoRE-Brain's fMRI routers to generalize it to the other 7 subjects, see Figure 9. We first train the model from scratch based on subject 1, then finetune only the fMRI routers in each level of MoE. The time and space routers trained based on subject 1 are directly used during inference on other subjects.

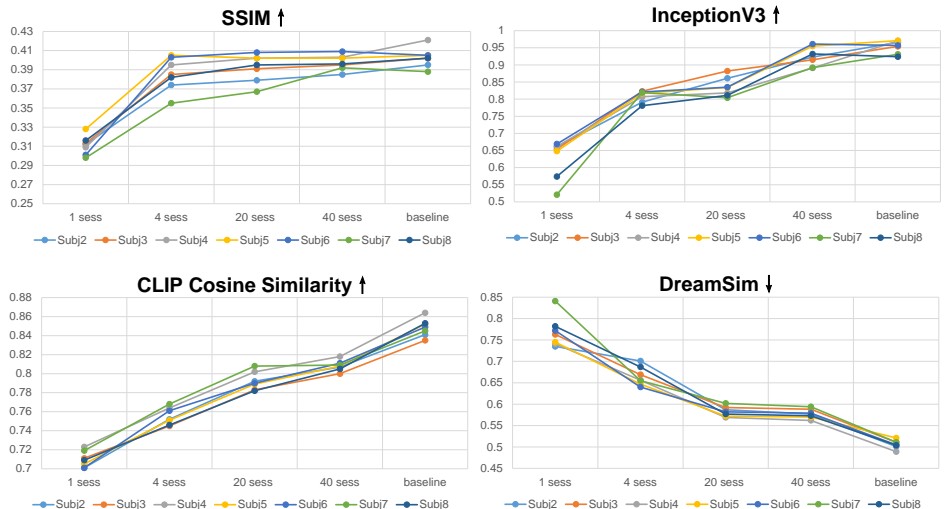

Figure 9: Generalize MoRE-Brain to other 7 subjects by finetuning only fMRI routers using different amounts of data. "baseline" denotes to retrain the model from scratch for the new subject.

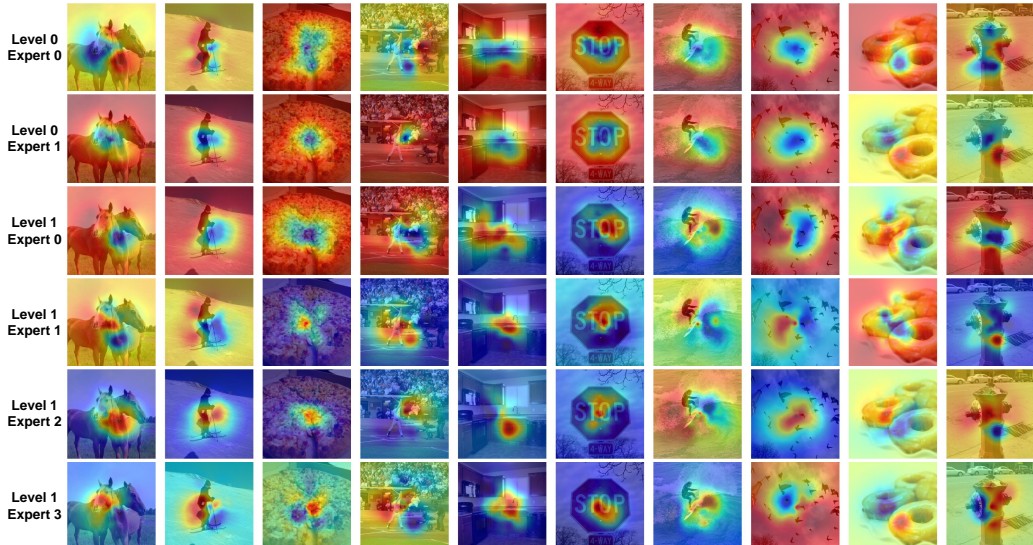

Figure 10: Predicted visual saliency maps for experts at MoE levels 0 and 1. Brighter regions indicate higher predicted importance to the expert. Saliency maps often focus on primary objects. Differences between experts at the same level (e.g., between $E_0^{(0)}$ and $E_1^{(0)}$ for the same image) highlight their specialized visual field focus.

# D   Image Saliency Predicted by MoRE-Brain

To visualize which parts of the input image are most influential for individual experts within MoRE-Brain, we project the voxel-space attribution scores for each expert onto the pixel space of the stimulus image. This transformation leverages the population receptive field (pRF) parameters provided by the NSD dataset. Each pRF model characterizes a voxel's response as a 2D Gaussian sensitivity profile in the visual field, defined by its center coordinates $(x_0, y_0)$ and size (standard deviation $\sigma$). For each voxel, its attribution value (calculated by GradientSHAP) is distributed onto a pixel grid representing the visual field. This distribution is weighted by the voxel's pRF Gaussian profile: the attribution is maximal at the pRF center and decreases with distance. The final saliency value for each pixel in the image is the sum of these weighted attributions from all voxels that have a pRF overlapping that pixel. This process effectively translates the expert-specific neural importance patterns from the fMRI voxel space back into a human-interpretable visual saliency map.

Figure 10 illustrates these predicted visual saliency maps for individual experts at the first two levels of the MoE hierarchy. The resulting saliency maps often highlight regions corresponding to the main object(s) or salient parts of the input image. Notably, while the saliency fields of different experts within the same level often overlap (as expected, since they are processing the same overall stimulus), they also exhibit clear divergences. These differences suggest that individual experts, even at the same hierarchical level, are learning to focus on distinct aspects or spatial regions of the visual input, driven by the different underlying neural populations they are routed from in the visual cortex. This supports the notion of functional specialization among experts.

# E   Reconstructions from a Single Level of Experts

To investigate the nature of information captured at different hierarchical levels of the MoE fMRI encoder, we conducted an experiment where image reconstructions were generated using only the expert embeddings from a single, specific level. For this analysis, the embeddings from all experts $E_j^{(l)}$ within a chosen level $l$ (i.e., $C^{(l)} = \{(c_{l,j}^{img}, c_{l,j}^{text})\}_{j=1}^{e_l}$) were averaged to form a single pair of image and text conditioning vectors for the SDXL model. The Time and Space Routers were bypassed in this setup.

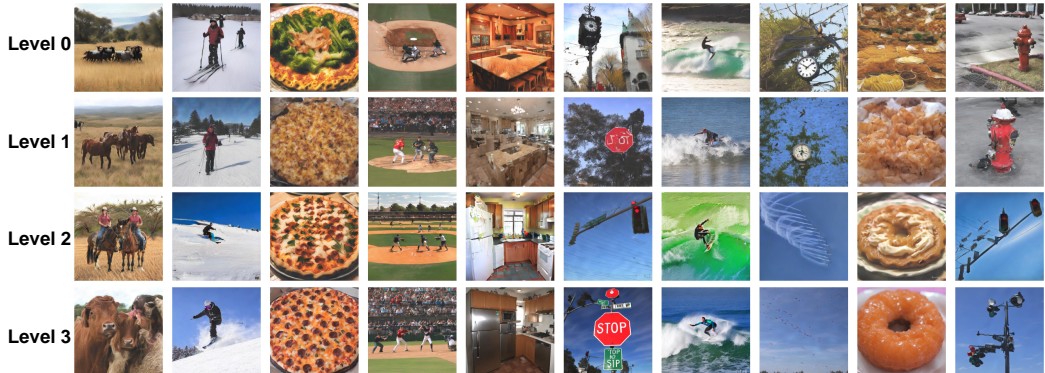

Figure 11: Qualitative examples of image reconstructions conditioned on expert embeddings from a single MoE level. Lower levels (coarse granularity) create reconstructions that may have wrong semantics, while higher levels reconstruct images more accurately.

## F    Analysis of the Time and Space Routers

### F.1    Expert Utilization

The Space Router ($\mathcal{R}_\mathcal{S}$) dynamically integrates embeddings from the experts of a selected MoE level to condition the SDXL U-Net (Eq. 6). Ideally, for a given selected level $l^*$, the Space Router should effectively utilize the information from all $e_{l^*}$ experts at that level, or at least show a balanced pattern of utilization if all experts are expected to contribute.

To assess this, we analyzed the attention weights assigned by the Space Router to the embeddings of individual experts within the selected level $l^*$. Specifically, in the cross-attention mechanism of the Space Router (where $z_t$ is query and $E_{sel}$ are keys/values), the attention scores over the $e_{l^*}$ experts indicate their contribution to the final conditioning $C_{fMRI}$. We averaged these attention scores for each expert across all diffusion timesteps and all test samples for which its level was selected by the Time Router.

Figure 12 displays the average utilization (mean attention score) for each expert within each of the $L = 4$ MoE levels. Within most levels, expert utilization is relatively balanced, suggesting that the Space Router generally draws information from all available experts at the selected level. Furthermore, some experts exhibit slightly higher average utilization (e.g., experts 2 and 3 in level 1). This could indicate that these experts either capture slightly more consistently relevant information or that the router develops a slight preference.

### F.2    Granularity Selection by the Time Router

The Time Router ($\mathcal{R}_\mathcal{T}$) is designed to select the most relevant MoE hierarchical level (granularity of fMRI features) at each diffusion timestep $t$, potentially mirroring coarse-to-fine visual processing dynamics. We analyzed the behavior of the Time Router when configured for *soft selection*, where it outputs attention weights $P_T \in \mathbb{R}^L$ over the $L$ levels (Equation 4). These weights indicate the preference for each level at a given timestep.

Figure 13 shows these attention scores $P_T$, averaged across all test images, for each MoE level as a function of the normalized diffusion timestep ($t/T$). The plot reveals a clear temporal dynamic where the router assigns higher weights for lower levels at early timesteps, and shifts its preference to higher levels at late timesteps. The Time Router learns to prioritize global scene structure and semantics encoded by lower-level experts during the initial stages of image generation, and then progressively incorporates finer details from higher-level experts as the generation process refines the image.

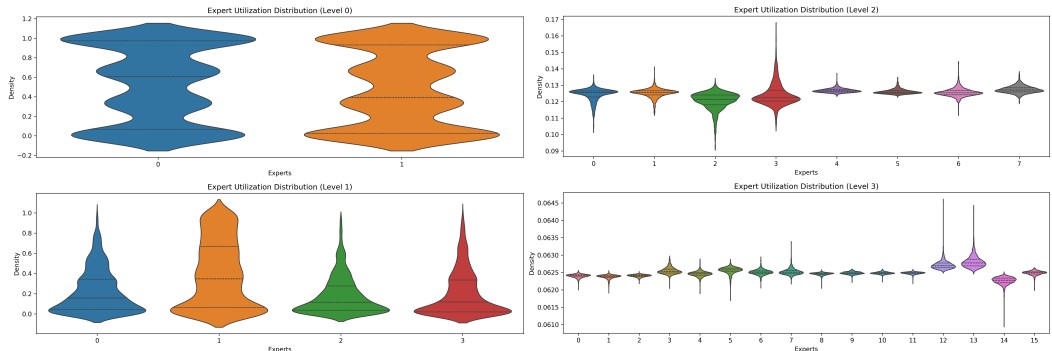

Figure 12: Average utilization of experts within each of the 4 MoE levels by the Space Router. Utilization is measured as the mean attention score assigned to an expert's embedding when its level is selected. Experts within each level generally show balanced utilization, suggesting effective integration of information from all specialized subnetworks.

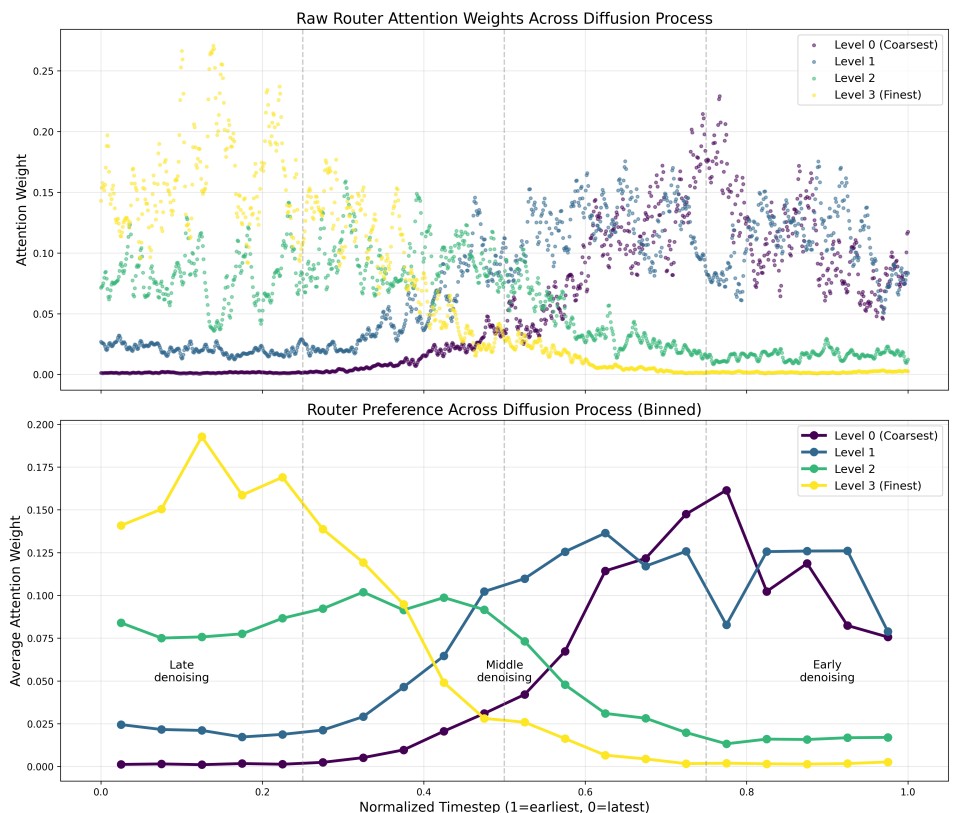

Figure 13: Time Router's learned preference (average attention weights $P_T$) for MoE granularity levels across normalized diffusion timesteps ($t/T$). Level 0 (coarsest) to Level 3 (finest). The router exhibits a clear coarse-to-fine selection pattern: lower levels (L0, L1) are preferred during early diffusion timesteps (left side of plot), while higher levels (L2, L3) are increasingly preferred during later timesteps (right side), guiding the generation from global structure to fine details.

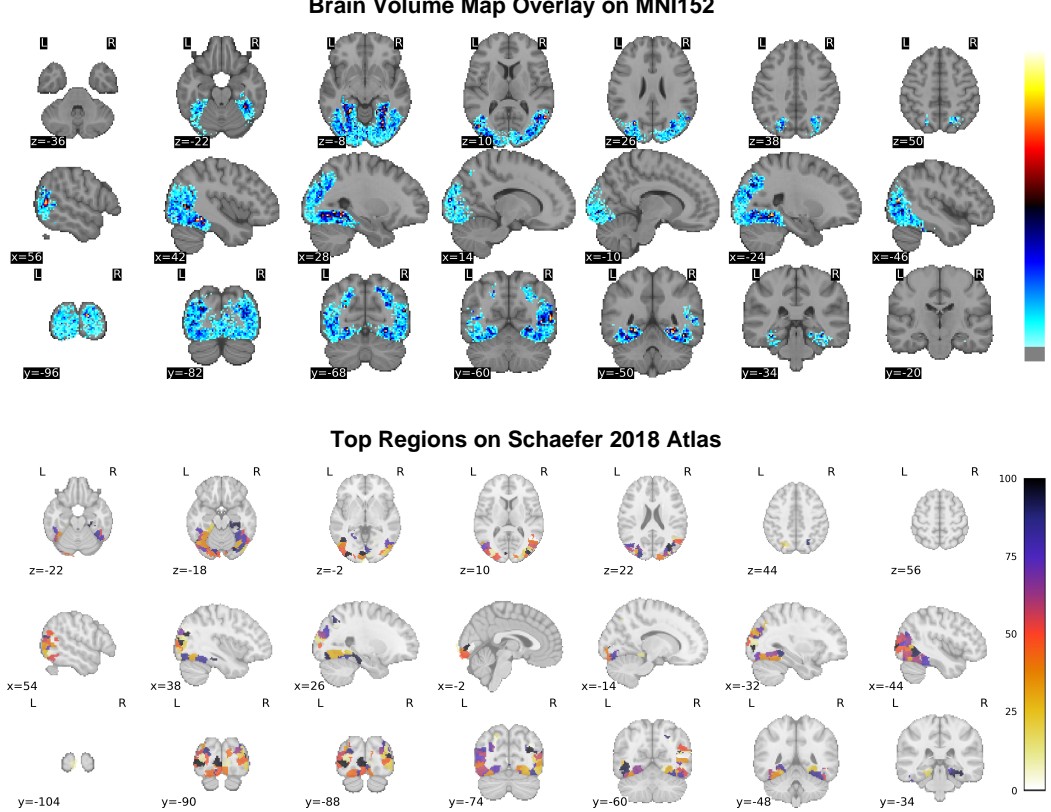

Figure 14: Comprehensive 2D slice visualizations for all 32 Independent Components (ICs) derived from GradientSHAP attributions, corresponding to Fig. 5.

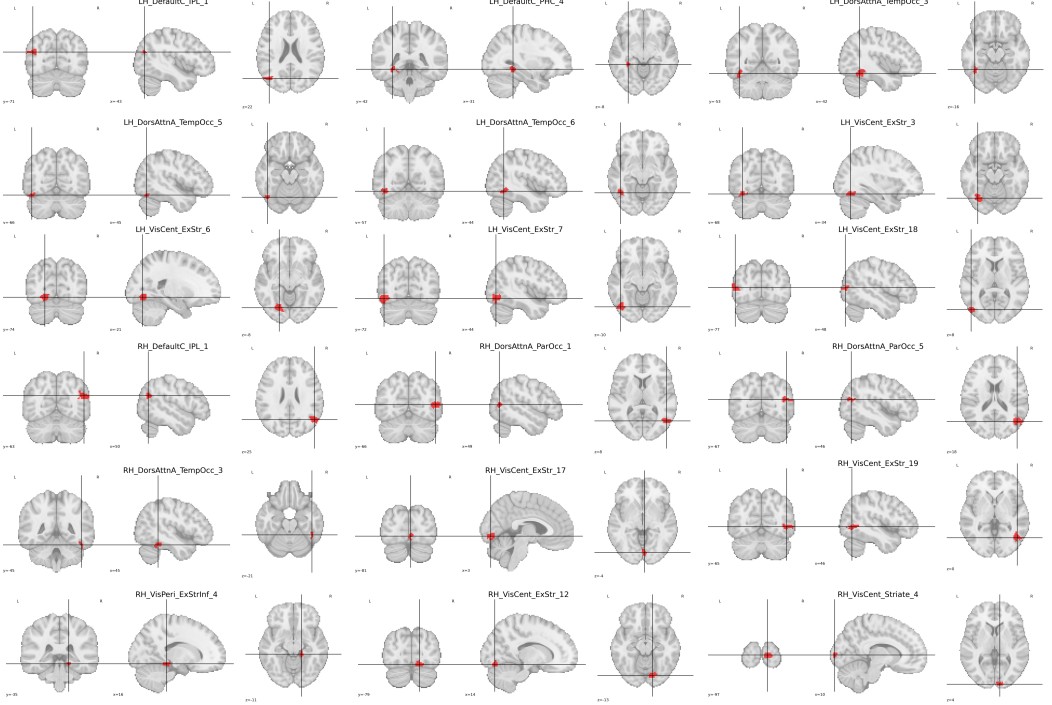

Figure 15: Top functional brain regions (Schaefer 2018 atlas, 1000 ROIs) associated with prominent Independent Components (ICs).

Table 4: Schaefer 2018 atlas label components and abbreviations

| Type | Abbreviation | Explanation |
|---|---|---|
| **Hemisphere** | | |
| | LH | Left Hemisphere |
| | RH | Right Hemisphere |
| **Network Name** | | |
| | Default"X" | Default Mode Network X (a sub-network of the DMN) |
| | DorsAttn"X" | Dorsal Attention Network X (a sub-network of the DAN) |
| | VisCent | Central Visual Network |
| | VisPeri | Peripheral Visual Network |
| **Anatomical Landmark/Region** | | |
| | IPL | Inferior Parietal Lobule |
| | PHC | Parahippocampal Cortex |
| | ParOcc | Parietal-Occipital region |
| | TempOcc | Temporo-Occipital region (junction of temporal and occipital lobes) |
| | ExStr | Extrastriate Cortex (visual cortex areas outside the primary visual cortex) |
| | Striate | Striate Cortex (Primary Visual Cortex, V1) |
| | StriCal | Striate and Calcarine Cortex (V1 area around the calcarine sulcus) |
| | ExStrInf | Inferior Extrastriate Cortex |

# G   ROIs for MoRE-Brain

We further show the complete 2D slices of Figure. 5 (see Figure 14 and the top regions labelled by Schaefer 2018 atlas that contribute to the reconstruction (see Figure 15. Explanations of abbreviations are in Table 4.

# H   Inter-Subject Variability in ICA Component Expression

The Independent Component Analysis (ICA) performed on the combined GradientSHAP attribution maps yields a mixing matrix. This matrix quantifies the extent to which each Independent Component (IC) is expressed in the attribution map of each specific image for each subject. To investigate inter-subject variability in how these neuro-computational patterns (represented by ICs) are utilized, we calculated the average mixing coefficient for each of the 32 ICs, separately for each of the 8 NSD subjects.

Figure 16 illustrates these subject-specific average IC expressions. The heatmap reveals significant inter-subject variations. For instance, Subject 1 shows strong average expression of IC15, IC17, and IC27, whereas Subject 3 strongly expresses IC8. These differences suggest that while the underlying computational components (ICs) identified by MoRE-Brain may be common, their degree of engagement or prominence can vary across individuals during the visual decoding task. This aligns with known inter-individual differences in brain function and anatomy and underscores the importance of subject-specific adaptation, which MoRE-Brain addresses through its router mechanism.

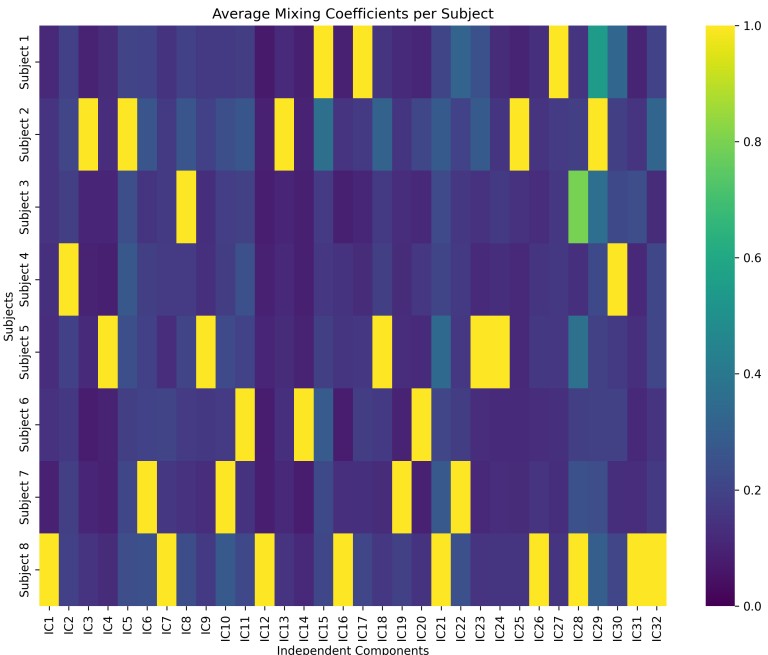

Figure 16: Inter-subject variability in the average expression of 32 Independent Components (ICs). Each cell represents the mean mixing coefficient of an IC (columns) for a given subject (rows). The distinct patterns across rows highlight that subjects differ in their reliance on or expression of these common functional components identified from fMRI attributions, motivating subject-specific model adaptation.

# I  Differential Brain Activity for Paired Semantic Categories

To further explore how MoRE-Brain utilizes different brain regions for processing distinct semantic content, we analyzed the ICs most strongly associated with specific pairs of image categories. For each category in a pair, we identified the top 3 ICs based on their average mixing coefficients for images belonging to that category. These ICs were then mapped to the Schaefer 2018 atlas to identify the corresponding functional brain regions. We selected three pairs for comparison: "electronic" vs. "vehicle", "animal" vs. "person", and "outdoor" vs. "sport", with the last pair being more semantically similar than the first two.

## I.1 Electronic vs. Vehicle

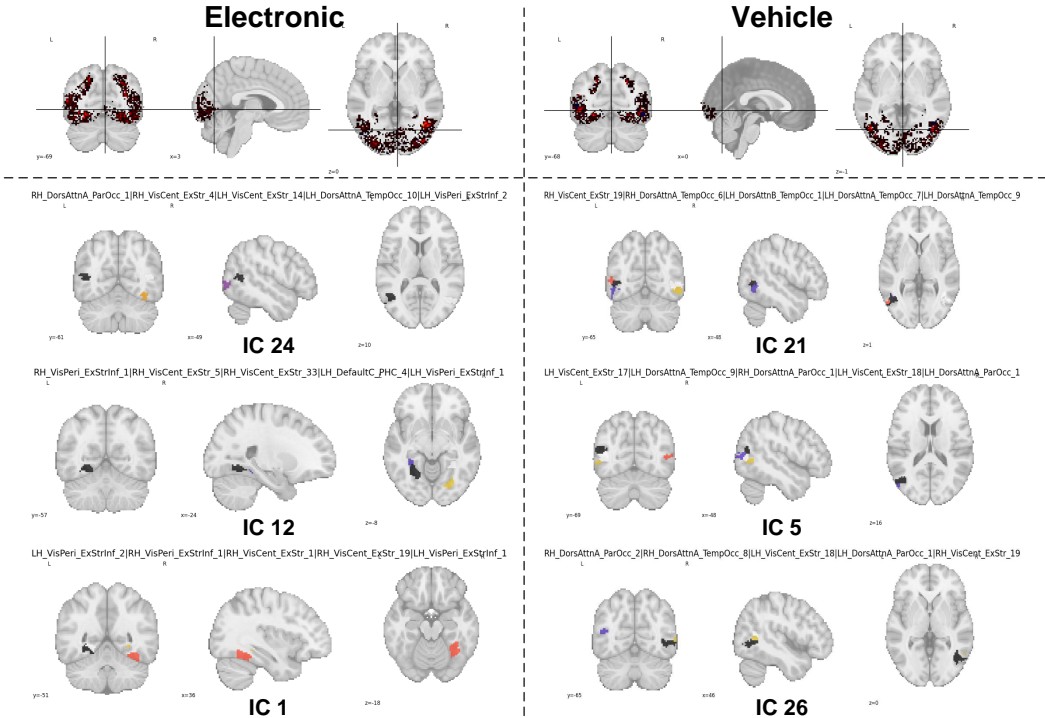

Figure 17: Comparison of top contributing Independent Components (ICs) and associated brain regions (Schaefer 2018 atlas) for "electronic" (left panels) versus "vehicle" (right panels) stimuli. Electronic objects significantly engage visual pathways (VisCent, VisPeri) for detailed analysis, attentional networks (DorsAttn), and memory-related regions like the Parahippocampal Cortex (PHC within Default Mode Network), potentially reflecting detailed feature encoding and retrieval of functional knowledge. Vehicle stimuli also heavily recruit visual and attentional networks, but are particularly distinguished by widespread and strong activation of the Dorsal Attention Network (DorsAttnA/B), suggesting a greater emphasis on spatial processing, implied motion, and interaction with the environment.

## I.2 Animal vs. Person

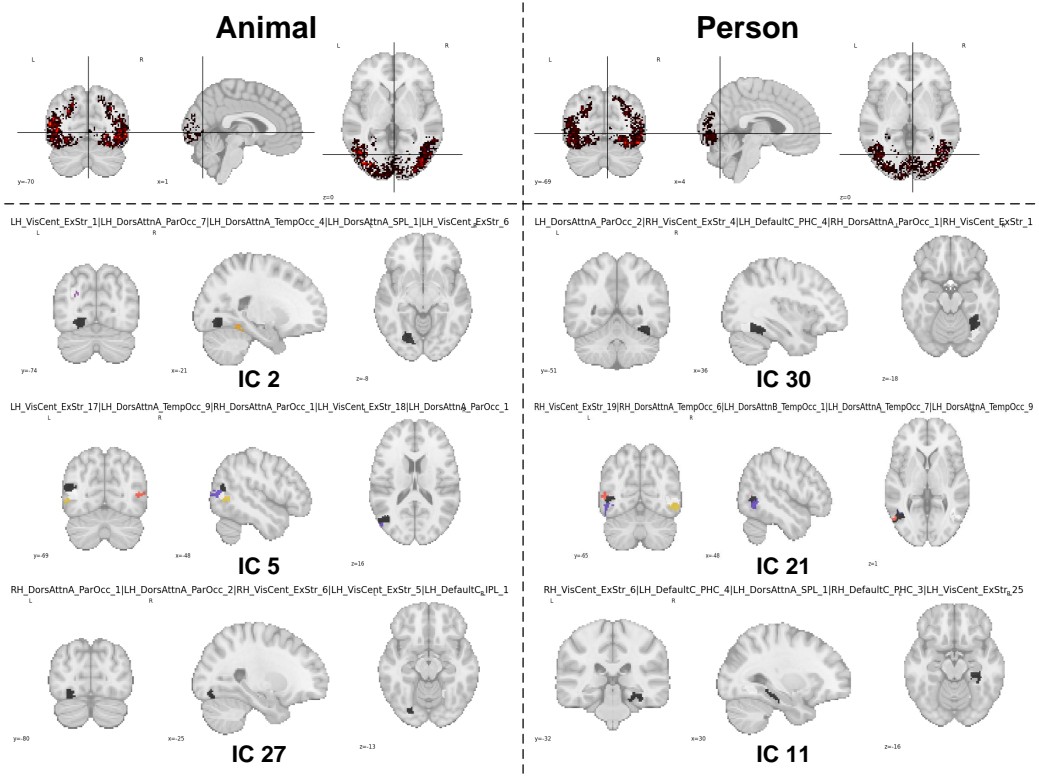

Figure 18: Comparison of top contributing ICs and associated brain regions for "animal" (left panels) versus "person" (right panels) stimuli. Both categories robustly activate extrastriate visual areas (e.g., VisCent_ExStr). Processing of person images shows more prominent involvement of Parahippocampal Cortex regions (DefaultC_PHC) within the Default Mode Network, potentially indicating enhanced episodic memory retrieval and social cognitive processes. Animal images also engage the Default Mode Network, notably via regions like the Inferior Parietal Lobule (DefaultC_IPL), possibly related to contextual association or theory of mind precursors.

## I.3 Outdoor vs. Sport

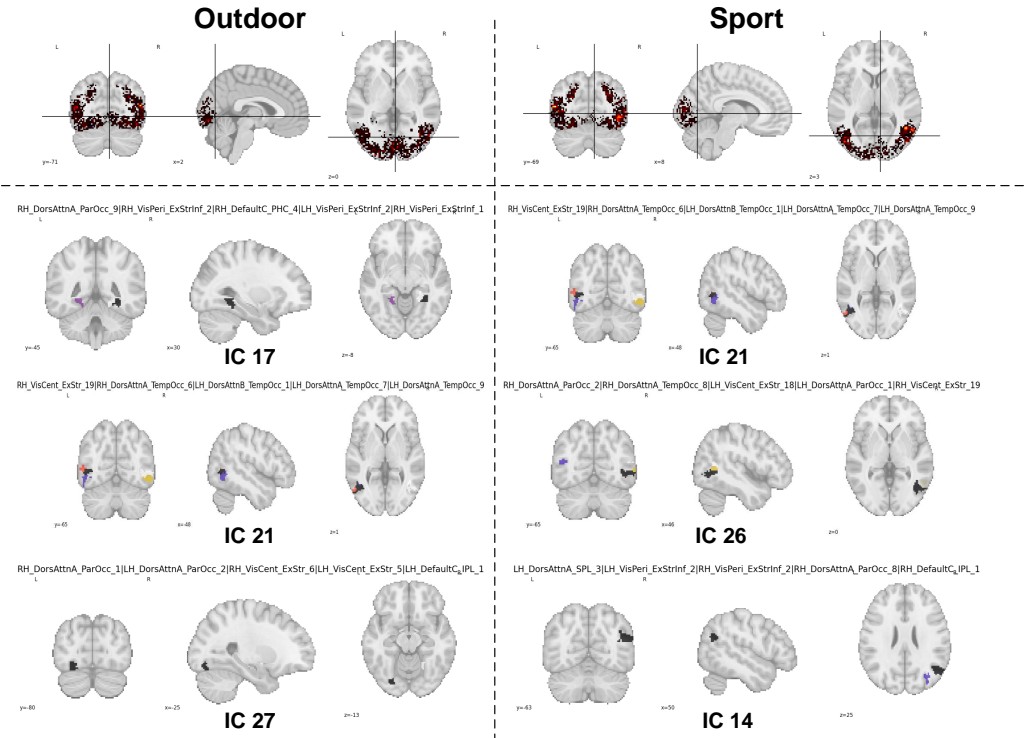

Figure 19: Comparison of top contributing ICs and associated brain regions for "outdoor" (left panels) versus "sport" (right panels) stimuli. These semantically related categories show considerable overlap in their neural correlates. Both robustly engage: (1) Visual networks (VisCent_ExStr, VisPeri_ExStrInf) for detailed scene analysis; (2) The Dorsal Attention Network (DorsAttnA, DorsAttnB) for processing spatial layouts and actual or implied actions; and (3) Regions of the Default Mode Network (e.g., DefaultC_IPL) likely involved in contextual understanding and memory associations. Subtle differences may exist in the relative weighting of these networks.

## J Hierarchical Functional Specialization of MoE Experts

We further provides detailed visualizations and descriptions of the functional specializations learned by the experts at each level $l$ of MoRE-Brain's hierarchical MoE fMRI encoder. The analyses are based on voxel attribution maps for each expert, mapped to the Schaefer 2018 atlas.To reflect the varying granularity and potential information integration across levels, we visualize a different number of top contributing ROIs for experts at different levels (e.g., top 8 ROIs for level 0, progressively fewer for higher levels, down to top 1 for level 3).

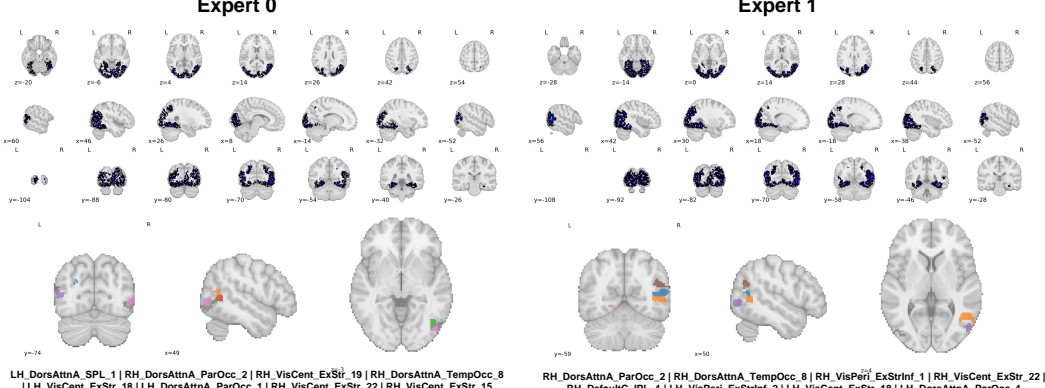

**Expert 0**

LH_DorsAttnA_SPL_1 | RH_DorsAttnA_ParOcc_2 | RH_VisCent_ExStr_19 | RH_DorsAttnA_TempOcc_8 | LH_VisCent_ExStr_18 | LH_DorsAttnA_ParOcc_1 | RH_VisCent_ExStr_22 | RH_VisCent_ExStr_15

**Expert 1**

RH_DorsAttnA_ParOcc_2 | RH_DorsAttnA_TempOcc_8 | RH_VisPeri_ExStrInf_1 | RH_VisCent_ExStr_22 | RH_DefaultC_IPL_1 | LH_VisPeri_ExStrInf_2 | LH_VisCent_ExStr_18 | LH_DorsAttnA_ParOcc_1

Figure 20: Functional specializations of experts at level 0. Experts at this initial level draw broadly from the visual system. Both experts show contributions from regions within the bilateral Dorsal Attention Network and the Visual Central Network, indicating processing across the visual hierarchy. $E_1^{(0)}$ additionally utilizes regions in the bilateral Visual Perisylvian Network, suggesting a possible emphasis on broader spatial orienting or integration with internally-guided attention alongside core visual processing.

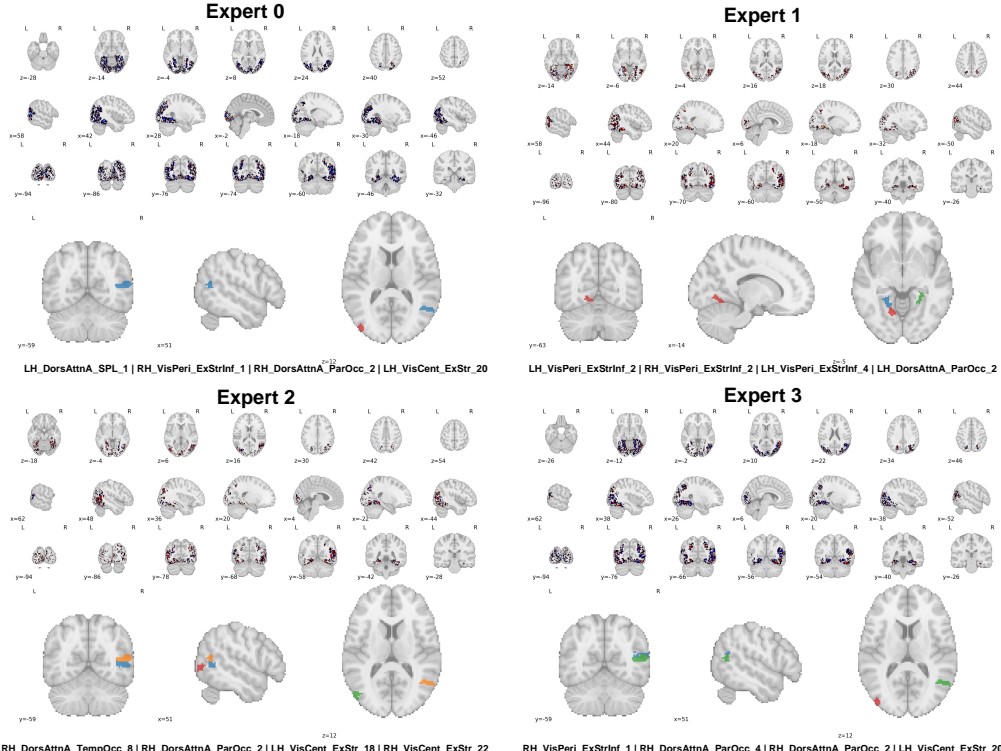

**Expert 0**

LH_DorsAttnA_SPL_1 | RH_VisPeri_ExStrInf_1 | RH_DorsAttnA_ParOcc_2 | LH_VisCent_ExStr_20

**Expert 1**

LH_VisPeri_ExStrInf_2 | RH_VisPeri_ExStrInf_2 | LH_VisPeri_ExStrInf_4 | LH_DorsAttnA_ParOcc_2

**Expert 2**

RH_DorsAttnA_TempOcc_8 | RH_DorsAttnA_ParOcc_2 | LH_VisCent_ExStr_18 | RH_VisCent_ExStr_22

**Expert 3**

RH_VisPeri_ExStrInf_1 | RH_DorsAttnA_ParOcc_4 | RH_DorsAttnA_ParOcc_2 | LH_VisCent_ExStr_20

Figure 21: Functional specializations of experts at level 1. Level 1 experts exhibit clearer divergence compared to level 0. For example, $E_1^{(1)}$ appears to specialize in processing signals from dorsal stream areas (spatial/motion, 'where' pathway), while $E_2^{(1)}$ may focus more on ventral stream areas (object recognition, 'what' pathway), potentially with hemispheric biases. Other experts like $E_0^{(1)}$ and $E_3^{(1)}$ might integrate information from both streams under attentional control, possibly with differing hemispheric emphases (e.g., $E_0^{(1)}$ recruiting bilateral attention including left SPL, $E_3^{(1)}$ showing strong right hemisphere Dorsal Attention involvement).

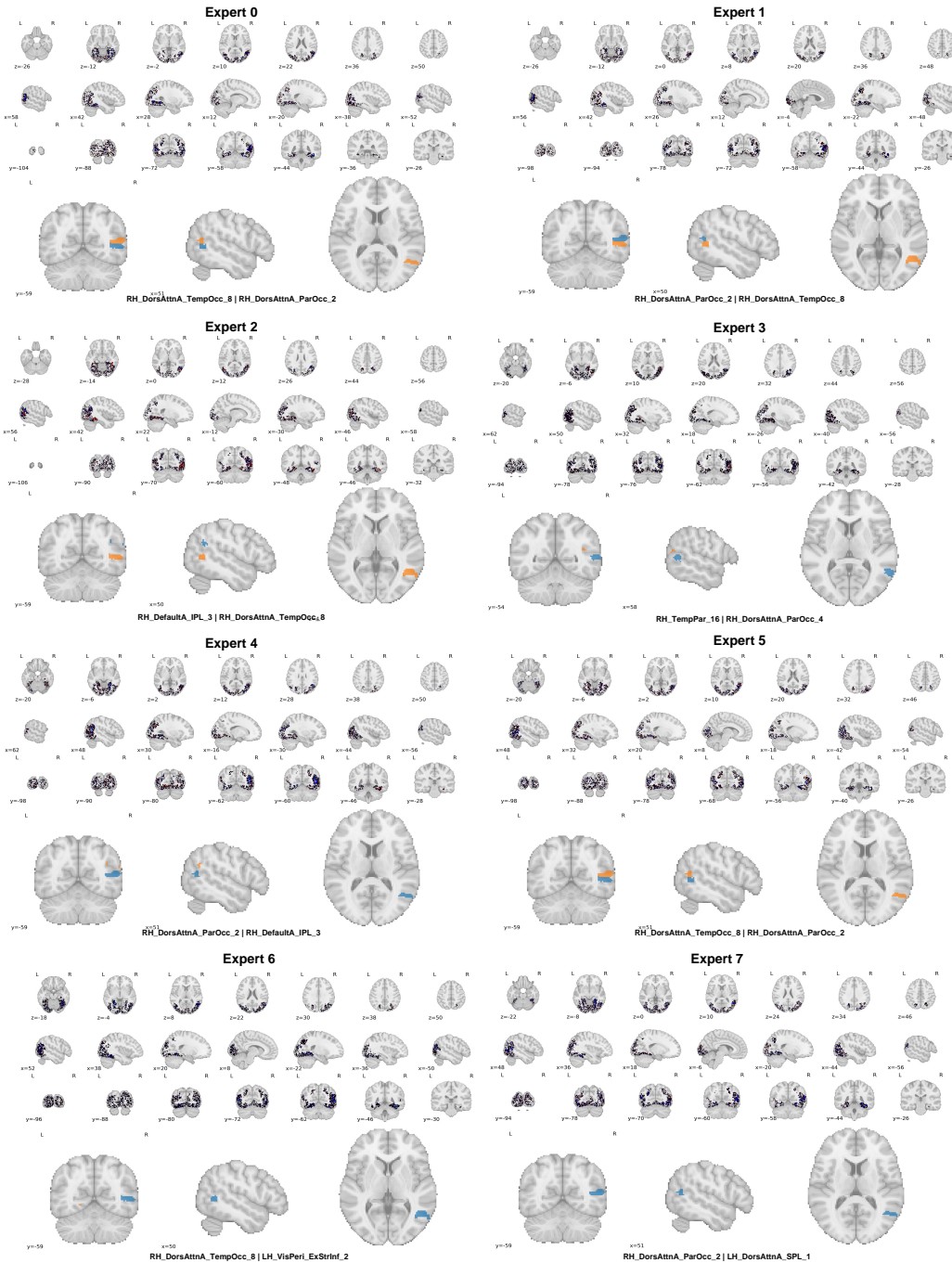

Figure 22: Functional specializations of experts at level 2. Experts appear to refine the specializations from level 1, applying more nuanced and context-dependent attentional modulation. For instance, specific experts might focus on outputs from 'what' or 'where' pathway specialists from L1 and further modulate these based on internal cognitive factors or more complex attentional strategies. This level could represent a stage of integrating specialized visual information with broader cognitive contexts.

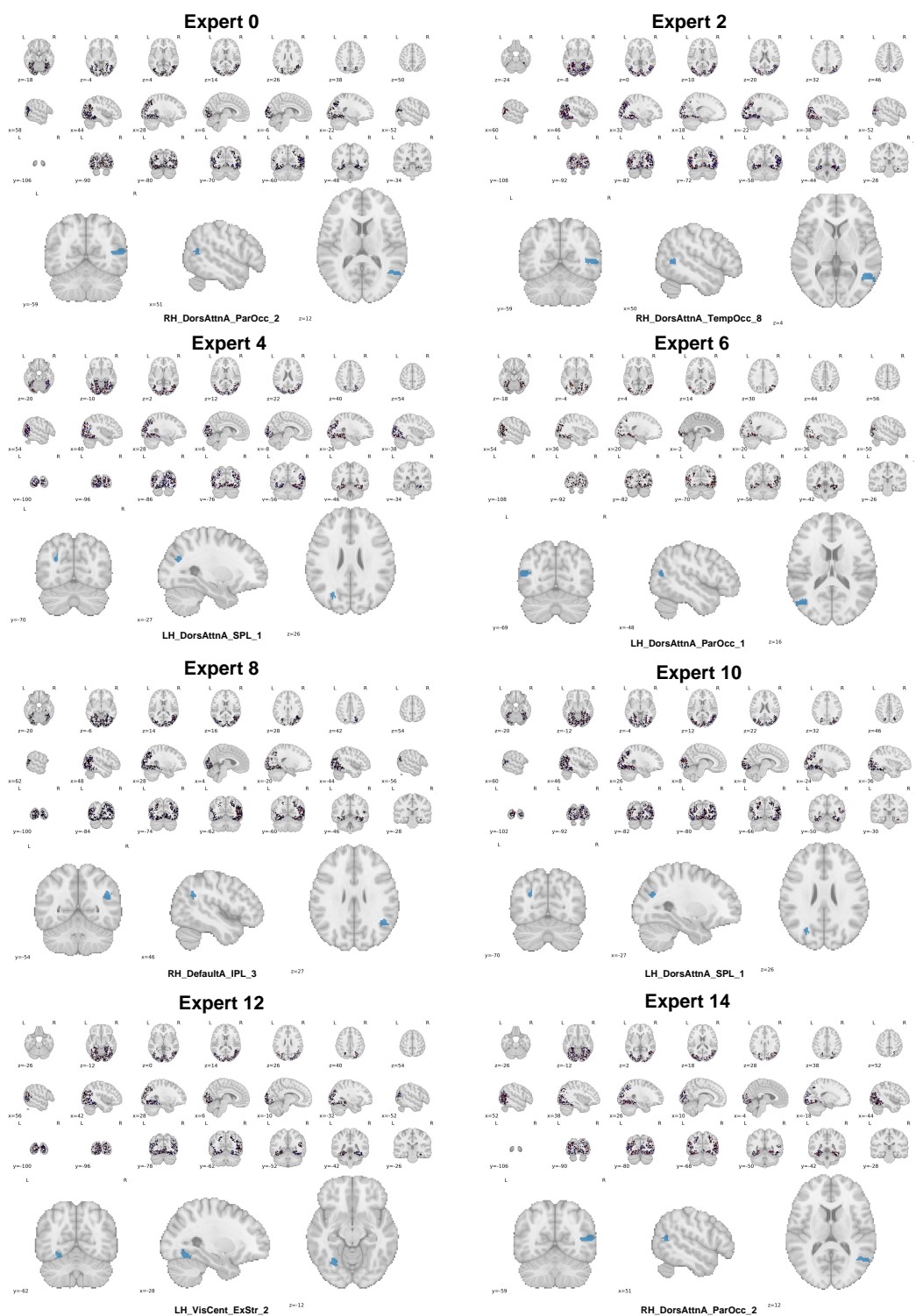

Figure 23: Functional specializations of experts at level 3. Experts demonstrate fine-grained specialization. The depicted experts show a distinct fractionation of the Dorsal Attention Network into specific regional and hemispheric components, suggesting focus on precise aspects such as spatial orienting (e.g., RH_DorsAttnA_ParOcc_2; LH_DorsAttnA_ParOcc_1), higher-order visual attention and integration (e.g., RH_DorsAttnA_TempOcc_8), and top-down attentional control or spatial working memory (e.g., LH_DorsAttnA_SPL_1). This highlights the emergence of highly specialized processing units at the apex of the hierarchy.

# K Limitations

While MoRE-Brain demonstrates significant advances in interpretable and generalizable fMRI decoding, several limitations should be acknowledged.

First, our approach relies on fMRI BOLD signals processed using GLM-Single to estimate activity betas for each stimulus presentation. This methodology inherently averages neural activity over the duration of the hemodynamic response, thus omitting the fine-grained temporal dynamics present in the raw fMRI time series and potentially faster neural processes. Consequently, MoRE-Brain decodes a relatively static representation of neural activity associated with a stimulus.

Second, the primary input features are derived from voxels within the 'nsdgeneral' ROI, which predominantly captures visually responsive cortex while potentially excluding contributions from other functional networks involved in higher-level cognition, memory retrieval, or contextual modulation that might also shape perception. As a result, MoRE-Brain is primarily optimized to decode the visual content directly driven by the stimulus ("what we see") rather than potentially more complex or internally generated perceptual states ("what we think" or imagine).

Finally, while our bottleneck analyses demonstrate that MoRE-Brain relies on actual neural signals rather than generative priors, we still use a diffusion model as in previous works to generate images. This potentially restricts us from correlating brain regions with spatial details of the image. Training an autoregressive model from scratch and directly mapping voxels to pixels may further enhance our understanding of the brain visual system.

