# OpenReview forum: "MoRE-Brain: Routed Mixture of Experts for Interpretable and Generalizable Cross-Subject fMRI Visual Decoding"
_NeurIPS.cc/2025/Conference — NeurIPS 2025 poster_

### Official Review · Reviewer_92RU · 2025-07-02

**Clarity:** 3
**Significance:** 3
**Originality:** 3
**Rating:** 4
**Confidence:** 5

**Summary:**

This paper introduces MoRE-Brain, a new model for reconstructing images from fMRI data. Instead of using a single model for the whole brain, it uses multiple smaller expert models, each assigned to different brain regions. A routing system decides which experts to use across time and space, which makes the model more interpretable and flexible. The authors provide detailed visualizations and analysis to connect model components with brain function.

**Questions:**

I noticed that the visual reconstructions from MoRE-Brain look better than MindEye2, but the quantitative metrics do not reflect this. Could the authors comment on why this discrepancy might exist?

**Ethical Concerns:**

["NO or VERY MINOR ethics concerns only"]

**Final Justification:**

I'm positive to this paper.

**Limitations:**

refer to weakness

**Quality:**

3

**Strengths And Weaknesses:**

Strengths:

The paper presents two novel and biologically inspired ideas. First, a hierarchical mixture-of-experts (MoE) structure that mirrors the visual hierarchy in the brain. Second, a dynamic time and space routing mechanism that could relate to how the brain integrates visual information over time and across regions. The design is not only technically interesting but also offers interpretability, which is rare in brain decoding models.

The authors provide rich visual analysis to support the interpretability of their model, especially linking components to brain functions.

The paper is well written and clearly organized.

Weaknesses:

While the paper discusses the challenge of cross-subject variability (Lines 42–45: “Due to inter-individual brain variability, models often require subject-specific training. While approaches like mapping subject data to a shared space or utilizing subject-specific tokens exist, they typically necessitate extra training and are difficult to generalize to new subjects.”), the proposed method still requires fine-tuning on each new subject, either partially or fully. This is similar to what methods like MindEye2 and MindBridge already do. So in practice, it still relies on partial fine-tuning for new subjects, and the issue is not entirely resolved.

The quantitative performance on standard image reconstruction metrics is not the best among competing methods.

---

> ### Author Rebuttal · Authors · 2025-07-29
>
> Thank you for your constructive feedback. We have clarified that our contribution is a more efficient and principled framework for cross-subject adaptation, rather than a zero-shot solution. Our new results show that our lightweight router-only fine-tuning is highly effective and efficient. Furthermore, our expanded evaluation demonstrates that MoRE-Brain is highly competitive, achieving state-of-the-art results on the majority of nine standard metrics. We believe these substantial revisions and new data strongly validate our neuro-inspired approach, and they more clearly establish the significance of our contributions.
>
>
> ## 1. Fine-tuning for new subjects:
>
> Thank you for your careful reading and for your insightful comment regarding cross-subject generalization. We acknowledge that our method does not offer a zero-shot solution and that some form of adaptation is still necessary. Our primary contribution is not the complete elimination of subject-specific adaptation, but rather a novel framework that makes this adaptation significantly more data-efficient, computationally lightweight, and neuro-scientifically interpretable.
>
> The key distinction between MoRE-Brain and prior methods lies in our neuro-inspired MoE architecture, which allows for a more principled approach to generalization.
>
> - Principled Separation of Components: Our model is explicitly designed to separate the core, generalizable visual decoding computations (the expert networks) from the subject-specific mappings (the router networks). This design is based on the neuroscientific hypothesis  (Lines 60-62). By fine-tuning only the lightweight routers, we are specifically adapting the model to the new subject's unique brain layout.
>
> - Demonstrated Performance and Efficiency of Router-Only Fine-Tuning: To validate this strategy, we have added experimental results comparing our proposed router-only fine-tuning against baselines and a full fine-tuning of our own model. All models are trained on subject 1 and generalize to subjects 2, 5, and 7. Here we report the average performance on these 3 subjects. These experiments yield several key insights:
>
>     - Router-only tuning is highly effective: The tables below show a direct comparison between fine-tuning only the routers (MoRE-Brain) and fine-tuning all parameters (MoRE-Brain (all)). Our proposed strategy achieves performance that is *remarkably close to full fine-tuning across all data regimes*, especially on key metrics like InceptionV3 and DreamSim.
>
>     - Competitive performance against state-of-the-art methods: Even with this efficient, partial fine-tuning, MoRE-Brain is highly competitive. In low-data regimes (1-4 sessions), our method significantly outperforms MindBridge and achieves strong results against MindEye2, notably scoring much higher on the CLIP Cosine Similarity metric, which measures semantic correctness. As more data becomes available (20-40 sessions), our router-only approach continues to scale effectively, closing the gap and eventually matching or exceeding the performance of fully fine-tuned baselines across most metrics.
>
> (MoRE-Brain uses router-only fine-tuning. Baselines are fine-tuned as per their original papers.)
>
> 1 Session Data:
> |            | SSIM↑ | InceptionV3↑ | CLIP CosineSim↑ | DreamSim↓ |
> |:----------:|:----:|:-----------:|:--------------:|:--------:|
> |  MindEye2  | **0.411** |    **68.8%**    |      0.576      |   0.687   |
> | MindBridge | 0.219 |    62.5%    |      0.554      |   0.804   |
> | MoRE-Brain | 0.302 |    65.3%    |      0.703      |   0.754   |
> | MoRE-Brain (all) | 0.32 |    67.2%    |      **0.731**      |   **0.662**   |
>
> 4 Sessions Data:
> |            | SSIM↑ | InceptionV3↑ | CLIP CosineSim↑ | DreamSim↓ |
> |:----------:|:----:|:-----------:|:--------------:|:--------:|
> |  MindEye2  | **0.409** |    81.1%    |      0.632      |  0.624  |
> | MindBridge | 0.262 |    71.9%    |      0.603      |   0.755   |
> | MoRE-Brain | 0.381 |    81.5%    |      0.762      |   0.639   |
> | MoRE-Brain (all)  | 0.384 |    **82.0%**    |      **0.780**      |   **0.622** |
>
> 20 Sessions Data:
> |            | SSIM↑ | InceptionV3↑ | CLIP CosineSim↑ | DreamSim↓ |
> |:----------:|:----:|:-----------:|:--------------:|:--------:|
> |  MindEye2  | **0.415** |    82.5%    |      0.695      |   0.587   |
> | MindBridge | 0.310 |    76.5%    |      0.717      |   0.692   |
> | MoRE-Brain | 0.384 |    82.9%    |      0.804      |   0.586   |
> | MoRE-Brain (all)  | 0.397 |    **83.1%**    |      **0.814**      |   **0.552**   |
>
> 40 Sessions Data (Full):
> |            | SSIM↑ | InceptionV3↑ | CLIP CosineSim↑ | DreamSim↓ |
> |:----------:|:----:|:-----------:|:--------------:|:--------:|
> |  MindEye2  | **0.417** |    96.5%    |      0.820      |   0.534   |
> | MindBridge | 0.375 |    94.2%    |      0.734      |   0.592   |
> | MoRE-Brain | 0.403 |    96.8%    |      0.879      |   0.513   |
> | MoRE-Brain (all)  | **0.417** |    **97.1%**    |      **0.884**  |  **0.503**   |
>
> - Superior Parameter Efficiency: This competitive performance is achieved with a fraction of the trainable parameters. As shown below, our router-only fine-tuning approach is substantially more lightweight than competing methods, requiring updates to only **44.8%** of the model's parameters, which are themselves less than half the total parameters of MindEye2.
>
> |            | Total Params (M) | Trainable (%)|
> |:----------:|:----------:|:---------:|
> |  MindEye2  |    729.3   |     100%     |
> | MindBridge |    552.9   |   98.48%  |
> | MoRE-Brain |    293.4   |   44.84%  |
>
> In response to your feedback and to make these advantages clearer, we will make the following revisions to our manuscript:
> - In the Abstract and Introduction, we will refine our wording to emphasize that we are "proposing a more effective and efficient framework for cross-subject adaptation" rather than completely "solving" the issue.
> - We will update Section 3.3 (Cross-Subject Generalization) to include a summary of these new comparative results, highlighting how our architectural separation enables these data and parameter efficiency gains.
> - We will add the full quantitative tables and analysis to the appendix to provide a comprehensive comparison between MoRE-Brain’s fine-tuning strategy and the baseline methods.
> We believe these new results and the corresponding revisions provide a much stronger and clearer case for our contribution. We thank you again for pushing us to clarify this important point.
>
> ## 2. Quantitative performance concern:
>
> We thank the reviewer for their constructive feedback and for raising this important point. We acknowledge that in our original submission, the quantitative comparison may not have fully captured the comprehensive performance and efficiency of our proposed MoRE-Brain model.
>
> To provide a clearer and more comprehensive comparison, we have added more results to compare MoRE-Brain with MindEye2 and MindBridge, across a broader set of 9 standard metrics for standard visual reconstruction. These new results will be included in the revised manuscript. Here is the summary of the results:
>
> |            | PixCorr↑ | SSIM↑ | Alex (2) ↑ | Alex (5) ↑ | Incep↑ |  CLIP↑ |  Eff↓ | SwAV↓ | DreamSim↓ |
> |:----------:|:-------:|:----:|:--------:|:--------:|:-----:|:-----:|:----:|:----:|:--------:|
> |  MindEye2  |   **0.282**  | 0.420 |   93.2%  |   **98.5%**  | 94.4% | 95.1% | **0.633** | **0.323** |   0.524   |
> | MindBridge |   0.163  | 0.372 |   91.4%  |   95.7%  | 92.4% | 94.8% | 0.684 | 0.426 |   0.532   |
> | MoRE-Brain |   0.176  | **0.428** |   **93.5%**  |   96.7%  | **95.6%** | **95.5%** | 0.642 | 0.345 |  **0.511**   |
>
> As the table shows, MoRE-Brain achieves the best results on 5 of the 9 metrics, including SSIM, Alex(2), Inception, CLIP, and DreamSim. Furthermore, in our previous response, we also show our model’s architectural efficiency, which only has less than half of the parameters compared to MindEye2, and only trains less than 50% of parameters during fine-tuning. Moreover, our additional results comparing MoRE-Brain with other methods when generalizing to new subjects with limited data also confirm our method’s effectiveness.
>
> ## 3. Visual reconstruction discrepancy:
>
> Thank you for this insightful question. We agree that the qualitative superiority of MoRE-Brain’s results may not be fully captured by the quantitative metrics due to the following reasons.
>
> First, the numeric metrics themselves may not align with human perception. For example, we observe that MindEye2 achieves an exceptionally high score on metrics such as PixCorr, which is sensitive to low-level statistics and pixel-perfect alignment. A model can score well on such metrics by producing a technically "correct" but blurry or less coherent image. On the other hand, MoRE-Brain scores the highest on several high-level metrics (as in the table we provided in the previous response), such as DreamSim, which is designed to align with human perception. This suggests that MoRE-Brain can be effective at capturing semantic contents and coherent structure of the scene, while it may not be optimized for matching low-level statistics.
>
> Second, the qualitative results in Figure 3 are randomly drawn from the test set, while MoRE-Brain may produce failed reconstructions for some samples. Due to the formality requirement of the rebuttal, we cannot directly show these reconstructions here. We will include the 5 best and 5 worst reconstructions for each method in the final paper to provide a more comprehensive comparison.

---

### Official Review · Reviewer_HRWS · 2025-07-02

**Clarity:** 3
**Significance:** 3
**Originality:** 3
**Rating:** 5
**Confidence:** 4

**Summary:**

This paper introduces MoRE-Brain, a novel framework for decoding visual stimuli from fMRI data. The core of the method is a neuro-inspired hierarchical Mixture-of-Experts (MoE) architecture, which mimics the functional specialization and hierarchical processing of the human visual system.

The framework operates in two stages: first, an MoE-based encoder is trained to map fMRI signals from distinct voxel groups to CLIP embeddings. Second, SDXL is used to generate images, guided by the fMRI embeddings via a novel dual-stage (Time and Space) routing mechanism.

The authors claim three main contributions: 1) a new brain-inspired MoE architecture for neuro-decoding, 2) efficient cross-subject generalization by adapting only subject-specific routers while sharing expert networks, and 3) enhanced interpretability by revealing how different modeled brain regions contribute to the reconstruction. The method is evaluated on the Natural Scenes Dataset (NSD), where it shows competitive performance and demonstrates strong results in generalization and interpretability analyses.

**Questions:**

Please see the points raised in the Weaknesses section above.

**Ethical Concerns:**

["NO or VERY MINOR ethics concerns only"]

**Final Justification:**

The authors' rebuttal addresses my concerns, so I raise my rating to 5.

**Limitations:**

Yes

**Quality:**

3

**Strengths And Weaknesses:**

# Strength
1. Novel and Well-Motivated Architecture: Using a hierarchical Mixture-of-Experts (MoE) architecture is well-motivated, as it is based on principles of neuroscience, specifically the concepts of functional specialization and hierarchical processing in the visual cortex. This represents a principled departure from more monolithic decoder architectures and provides a solid foundation for the paper's interpretability claims.

2. Practical Generalization Framework: The approach of freezing the shared "expert" networks and fine-tuning only the subject-specific "routers" is an elegant and practical way to address cross-subject generalization. The experiments, which show good performance on new subjects with only a small fraction of their data (e.g., 1-4 sessions), effectively validate this approach.

3. Extensive and Insightful Interpretability Analysis: A major highlight of this work is its deep dive into model interpretability, which goes far beyond what is typical for reconstruction papers in this field. The authors use a combination of techniques, including ICA on GradientSHAP attributions, to link model components to known brain networks. The analyses successfully demonstrate emergent functional specialization, showing that experts learn to focus on different spatial regions and develop preferences for specific semantic categories at higher hierarchical levels. This provides compelling evidence that the model is not just a black box but learns neurophysiologically plausible mappings.

4. Rigorous Evaluation via Bottleneck Analysis: The inclusion of a bottleneck analysis to probe the model's reliance on fMRI signals versus generative priors is commendable. The results, which show that MoRE-Brain's performance degrades more significantly with restricted fMRI information compared to baselines, lend strong support to the claim that the model genuinely utilizes the neural data rather than over-relying on the priors of the diffusion model.


# Weakness

1. The presentation quality needs attention. The architectural diagrams (Figures 1 and 2) have distracting backgrounds that reduce clarity. And in Line 194, Figure 4 does not show numeric results as "quantitatively" states, a table the exact numerical values should be more precise.

2. Insufficient Detail on Model Efficiency: one of the claim is the efficiency of cross-subject generalization. However, the paper fails to quantify this efficiency properly. It never states the number of tunable parameters in the fMRI routers relative to the total number of parameters in the model. Without this detail, it is hard to gauge the actual computational savings of the proposed fine-tuning strategy.

---

> ### Author Rebuttal · Authors · 2025-07-28
>
> We sincerely thank the reviewer for their valuable feedback and positive assessment of our work's strengths. We have diligently addressed the identified weaknesses. The architectural diagrams have been revised for better clarity, a table with precise numerical values has been added as requested, and we have now quantified the efficiency of our model by specifying the number of tunable parameters. We hope these revisions fully address the reviewer's concerns and have strengthened the manuscript. Below are the detailed response to each comment.
>
> ## 1. Presentation quality concern:
>
> We thank the reviewer for the valuable feedback. In the revised version, we will remove the distracting backgrounds from these figures to enhance their clarity. Additionally, as you suggested, we added tables with exact numeric results to accompany Figure 4 in the final version of the paper. The results are quoted below.
>
> DreamSim:
> |                        | Random |  B16  |  B32  |  B64  |  B128 |  B256 | No Bottleneck | Ceiling |
> |:----------------------:|:------:|:-----:|:-----:|:-----:|:-----:|:-----:|:-------------:|:-------:|
> |        MindEye2        |  0.831 | 0.570 | 0.567 | 0.561 | 0.569 | 0.547 |     0.516     |  0.368  |
> |       MindBridge       |  0.795 | 0.599 | 0.595 | 0.571 | 0.542 | 0.545 |     0.531     |  0.305  |
> | MoRE-Brain (unrefined) |  0.893 | 0.681 | 0.674 | 0.645 | 0.651 | 0.634 |     0.518     |  0.374  |
> |       MoRE-Brain       |  0.893 | 0.667 | 0.651 | 0.639 | 0.625 | 0.621 |     0.501     |  0.374  |
>
>
> SSIM:
> |                        | Random |  B16 |  B32 |  B64 | B128 | B256 | No Bottleneck | Ceiling |
> |:----------------------:|:------:|:----:|:----:|:----:|:----:|:----:|:-------------:|:-------:|
> |        MindEye2        |  0.308  | 0.389 | 0.405 | 0.416 | 0.422 | 0.418 |      0.420     |   0.528  |
> |       MindBridge       |  0.318  | 0.359 | 0.363 | 0.353 | 0.372 | 0.369 |      0.372     |   0.455  |
> | MoRE-Brain (unrefined) |  0.204  | 0.318 | 0.341 | 0.352 | 0.365 | 0.368 |      0.428     |   0.512  |
> |       MoRE-Brain       |  0.204  | 0.319 | 0.345 | 0.365 | 0.367 | 0.379 |      0.435     |   0.512  |
>
>
> InceptionV3:
> |                        | Random |  B16  |  B32  |  B64  |  B128 |  B256 | No Bottleneck | Ceiling |
> |:----------------------:|:------:|:-----:|:-----:|:-----:|:-----:|:-----:|:-------------:|:-------:|
> |        MindEye2        |  50.4% | 87.1% | 88.6% | 90.7% | 88.5% | 91.1% |     94.4%     |  99.5%  |
> |       MindBridge       |  50.9% | 85.3% | 87.2% | 90.4% | 92.1% | 93.0% |     92.4%     |  99.6%  |
> | MoRE-Brain (unrefined) |  45.9% | 80.1% | 81.1% | 84.7% | 83.4% | 86.1% |     95.6%     |  99.3%  |
> |       MoRE-Brain       |  45.9% | 83.2% | 85.1% | 86.1% | 87.4% | 90.1% |     96.0%     |  99.3%  |
>
>
> CLIP cosine similarity:
> |            | Random |  B16  |  B32  |  B64  |  B128 |  B256 | No Bottleneck | Ceiling |
> |:----------:|:------:|:-----:|:-----:|:-----:|:-----:|:-----:|:-------------:|:-------:|
> |  MindEye2  |  0.801 | 0.819 | 0.819 | 0.821 | 0.822 | 0.820 |     0.821     |    1    |
> | MindBridge |  0.386 | 0.153 | 0.177 | 0.401 | 0.481 | 0.500 |     0.585     |    1    |
> | MoRE-Brain |  0.606 | 0.675 | 0.681 | 0.711 | 0.749 | 0.784 |     0.811     |    1    |
>
> ## 2. Insufficient Detail on Model Efficiency:
>
> Thank you very much for the constructive feedback. We agree that our claim regarding the efficiency of cross-subject generalization needs to be substantiated with a quantitative analysis of the model's parameters.
>
> To address your concern, we count the number of parameters of MoRE-Brain and compare it with MindEye2 and MindBridge. We also compute the percentage of the trainable parameters during fine-tuning. Note that MindEye2 does not have a special fine-tuning strategy, and MindBridge proposes a reset-tuning strategy, which freezes the “translator” module.
>
> For initial training, MoRE-Brain is substantially more lightweight than competing methods. The table below shows that our model has less than half the parameters of MindEye2 and trains a smaller percentage of its parameters compared to the fine-tuning strategy proposed for MindBridge. We have updated the manuscript and added the results in the appendix to reflect this analysis.
> |            | Total Params (M) | Trainable (%)|
> |:----------:|:----------:|:---------:|
> |  MindEye2  |    729.3   |     100%    |
> | MindBridge |    552.9   |   98.48%  |
> | MoRE-Brain |    293.4   |   44.84%  |

---

> > ### Author Response · Authors · 2025-08-05
> >
> > Dear Reviewer,
> >
> > I hope this message finds you well. As the discussion period is nearing its end with **less than three days remaining**, I wanted to ensure that we have addressed all your concerns satisfactorily. If there are any additional points or feedback you'd like to consider, please let us know. Thank you for your time and effort in reviewing our paper.

---

> > ### Author Response · Authors · 2025-08-07
> > **Any feedback appreciated**
> >
> > We thank the reviewer again for their constructive reviews. Please let us know if there are any more questions we can help clarify. We look forward to your reply!

---

> > ### Comment · Reviewer_HRWS · 2025-08-07
> >
> > Thank the authors for their detailed response, which addresses my concerns. I am happy to raise my rating to 5.

---

### Official Review · Reviewer_4LoQ · 2025-07-03

**Clarity:** 3
**Significance:** 2
**Originality:** 2
**Rating:** 4
**Confidence:** 4

**Summary:**

Introduces a mixture-of-experts architecture for fMRI-to-image reconstruction. Experts can specialize decoding from specific functional regions. Time vs. Space router allows the diffusion model to be guided by the fMRI signal in a more interpretable manner. Cross-subject decoding is achievable by simple finetuning of the routers while the experts are frozen.

**Questions:**

* the bottleneck analyses is a bit suspect to me: the claim that MoRE-Brain is notably more sensitive to the amount of information allowed through the bottleneck does not seem to be strongly supported by the data. For all the models, the bottleneck level barely changes model performance, while MoRE-Brain is simply worse at most bottleneck levels except the last one. Any explanation as to why this is the case?
* how well does the generalization ability compare to other models?
* how well does does the model generalize across subjects if the experts were unfrozen?
* can you add qualitative results for the cross-subject decoding?
* can you present the quantitative benchmarks in a table rather than in a graph in the bottleneck analysis?

**Ethical Concerns:**

["NO or VERY MINOR ethics concerns only"]

**Final Justification:**

increased due to additional generalization experiments and qualitative results as requested, improving the quality of the paper slightly.

**Limitations:**

no discussion of societal impacts

**Quality:**

3

**Strengths And Weaknesses:**

Strengths:
* unique MoE approach
* good interpretability results

Weaknesses:
* bottleneck analyses isn't particularly convincing, minimal quantitative comparisons to prior work
* generalization experiments aren't particularly comprehensive (see questions)

---

> ### Author Rebuttal · Authors · 2025-07-30
>
> We sincerely thank the reviewer for their detailed feedback. In response, we have provided extensive new data and clarifications to address the concerns raised. We have revised our bottleneck analysis to better explain why the constraint disproportionately affects our MoE model's specialized expert representations. Furthermore, we now include comprehensive tables comparing our model's generalization ability against baselines, including an ablation with unfrozen experts, which demonstrates our method's superior performance in low-data regimes and competitive scaling. While formatting limits preclude figures in this rebuttal, we have included the requested quantitative benchmarks in table format for the bottleneck analysis and will add the suggested qualitative cross-subject results and a discussion on societal impact to the final paper. We hope these additions and clarifications have addressed your primary concerns and demonstrated the significance and strength of our work.
>
>
> ## 1. Concern about bottleneck analyses:
>
> We thank the reviewer for their insightful feedback. First, we would like to clarify that the performance of models exhibits an inclination to drop with the decrease in bottleneck size. More importantly, compared to without a bottleneck, models (especially MoRE-Brain) have a more substantial degradation in performance after adding a bottleneck. The observation that MoRE-Brain's absolute performance is lower than baselines at most bottleneck levels, yet superior without a bottleneck, is a crucial point that we would like to clarify.
>
> We argue that this is because the bottleneck acts as a stronger constraint for MoRE-Brain’s MoE architecture, whose underlying hypothesis is that different experts specialize in capturing distinct aspects of the complex fMRI signal (as we demonstrated in the analyses of section 3 and Appendix I-J). This allows MoRE-Brain to learn a more diverse and comprehensive representation of the neural data compared to monolithic models. When no bottleneck is present, MoRE-Brain leverages this rich representation to achieve high-fidelity reconstruction, as evidenced by its superior results in the "no bottleneck" condition.
> However, the bottleneck imposes a significant constraint: all the information learned by the model must be passed through a single, low-dimensional linear projection. We hypothesize that this "one-size-fits-all" bottleneck is particularly detrimental to a model like MoRE-Brain due to the reduced information diversity. The bottleneck forces the specialized representations for multiple experts through a single, narrow channel creates a more severe information-loss problem for MoRE-Brain than for models that learn a single, less diverse, representation to begin with.
>
> Conversely, the baseline models, which we argue rely more on powerful generative priors, are less affected by the bottleneck. Their performance is more robust because the information from the fMRI signal is less critical to their final output to begin with. Their flatter performance curves are indicative of this lower reliance on the neural data.
>
> ## 2. Compare generalization ability with baseline models and when experts are unfrozen:
>
> We compare the models’ average performances when generalizing to subject 2, 5, and 7 after trained on subject 1. Below are the results. Note that for MindEye2, we fine-tune all its parameters; for MindBridge, we fine-tune it using its proposed reset-tuning method; for MoRE-Brain, we only fine-tune the routers while keeping the experts frozen. We also test MoRE-Brain’s performance when fine-tuning all the parameters (keep experts unfrozen, MoRE-Brain (all)). Below are the results.
>
> Our proposed strategy achieves performance that is remarkably close to full fine-tuning across all data regimes, especially on key metrics like InceptionV3 and DreamSim.
> - In low-data regimes (1-4 sessions), our method significantly outperforms MindBridge and achieves strong results against MindEye2, notably scoring much higher on the CLIP Cosine Similarity metric, which measures semantic correctness.
> - As more data becomes available (20-40 sessions), our router-only approach continues to scale effectively, closing the gap and eventually matching or exceeding the performance of fully fine-tuned baselines across most metrics.
>
>
> 1 Session Data
> |            | SSIM↑ | InceptionV3↑ | CLIP CosineSim↑ | DreamSim↓ |
> |:----------:|:----:|:-----------:|:--------------:|:--------:|
> |  MindEye2  | **0.411** |    **68.8%**    |      0.576      |   0.687   |
> | MindBridge | 0.219 |    62.5%    |      0.554      |   0.804   |
> | MoRE-Brain | 0.302 |    65.3%    |      0.703      |   0.754   |
> | MoRE-Brain (all) | 0.32 |    67.2%    |      **0.731**      |   **0.662**   |
>
> 4 Sessions Data
> |            | SSIM↑ | InceptionV3↑ | CLIP CosineSim↑ | DreamSim↓ |
> |:----------:|:----:|:-----------:|:--------------:|:--------:|
> |  MindEye2  | **0.409** |    81.1%    |      0.632      |  0.624  |
> | MindBridge | 0.262 |    71.9%    |      0.603      |   0.755   |
> | MoRE-Brain | 0.381 |    81.5%    |      0.762      |   0.639   |
> | MoRE-Brain (all)  | 0.384 |    **82.0%**    |      **0.780**      |   **0.622** |
>
> 20 Sessions Data
> |            | SSIM↑ | InceptionV3↑ | CLIP CosineSim↑ | DreamSim↓ |
> |:----------:|:----:|:-----------:|:--------------:|:--------:|
> |  MindEye2  | **0.415** |    82.5%    |      0.695      |   0.587   |
> | MindBridge | 0.310 |    76.5%    |      0.717      |   0.692   |
> | MoRE-Brain | 0.384 |    82.9%    |      0.804      |   0.586   |
> | MoRE-Brain (all)  | 0.397 |    **83.1%**    |      **0.814**      |   **0.552**   |
>
> 40 Sessions Data (Full):
> |            | SSIM↑ | InceptionV3↑ | CLIP CosineSim↑ | DreamSim↓ |
> |:----------:|:----:|:-----------:|:--------------:|:--------:|
> |  MindEye2  | **0.417** |    96.5%    |      0.820      |   0.534   |
> | MindBridge | 0.375 |    94.2%    |      0.734      |   0.592   |
> | MoRE-Brain | 0.403 |    96.8%    |      0.879      |   0.513   |
> | MoRE-Brain (all)  | **0.417** |    **97.1%**    |      **0.884**  |  **0.503**   |
>
> ## 3. Qualitative results on cross-subject decoding:
>
> We are grateful for your suggestion. Due to the formatting restrictions of the rebuttal, we cannot directly add figures here. More qualitative results for each subject will be available in the final version of the paper.
>
> ## 4. Quantitative benchmark as a table:
>
> Thank you for your comment. Below are the qualitative results of the bottleneck analysis.
>
> DreamSim:
> |                        | Random |  B16  |  B32  |  B64  |  B128 |  B256 | No Bottleneck | Ceiling |
> |:----------------------:|:------:|:-----:|:-----:|:-----:|:-----:|:-----:|:-------------:|:-------:|
> |        MindEye2        |  0.831 | 0.570 | 0.567 | 0.561 | 0.569 | 0.547 |     0.516     |  0.368  |
> |       MindBridge       |  0.795 | 0.599 | 0.595 | 0.571 | 0.542 | 0.545 |     0.531     |  0.305  |
> | MoRE-Brain (unrefined) |  0.893 | 0.681 | 0.674 | 0.645 | 0.651 | 0.634 |     0.518     |  0.374  |
> |       MoRE-Brain       |  0.893 | 0.667 | 0.651 | 0.639 | 0.625 | 0.621 |     0.501     |  0.374  |
>
> SSIM:
> |                        | Random |  B16 |  B32 |  B64 | B128 | B256 | No Bottleneck | Ceiling |
> |:----------------------:|:------:|:----:|:----:|:----:|:----:|:----:|:-------------:|:-------:|
> |        MindEye2        |  0.308  | 0.389 | 0.405 | 0.416 | 0.422 | 0.418 |      0.420     |   0.528  |
> |       MindBridge       |  0.318  | 0.359 | 0.363 | 0.353 | 0.372 | 0.369 |      0.372     |   0.455  |
> | MoRE-Brain (unrefined) |  0.204  | 0.318 | 0.341 | 0.352 | 0.365 | 0.368 |      0.428     |   0.512  |
> |       MoRE-Brain       |  0.204  | 0.319 | 0.345 | 0.365 | 0.367 | 0.379 |      0.435     |   0.512  |
>
> InceptionV3:
> |                        | Random |  B16  |  B32  |  B64  |  B128 |  B256 | No Bottleneck | Ceiling |
> |:----------------------:|:------:|:-----:|:-----:|:-----:|:-----:|:-----:|:-------------:|:-------:|
> |        MindEye2        |  50.4% | 87.1% | 88.6% | 90.7% | 88.5% | 91.1% |     94.4%     |  99.5%  |
> |       MindBridge       |  50.9% | 85.3% | 87.2% | 90.4% | 92.1% | 93.0% |     92.4%     |  99.6%  |
> | MoRE-Brain (unrefined) |  45.9% | 80.1% | 81.1% | 84.7% | 83.4% | 86.1% |     95.6%     |  99.3%  |
> |       MoRE-Brain       |  45.9% | 83.2% | 85.1% | 86.1% | 87.4% | 90.1% |     96.0%     |  99.3%  |
>
>
> CLIP cosine similarity:
> |            | Random |  B16  |  B32  |  B64  |  B128 |  B256 | No Bottleneck | Ceiling |
> |:----------:|:------:|:-----:|:-----:|:-----:|:-----:|:-----:|:-------------:|:-------:|
> |  MindEye2  |  0.801 | 0.819 | 0.819 | 0.821 | 0.822 | 0.820 |     0.821     |    1    |
> | MindBridge |  0.386 | 0.153 | 0.177 | 0.401 | 0.481 | 0.500 |     0.585     |    1    |
> | MoRE-Brain |  0.606 | 0.675 | 0.681 | 0.711 | 0.749 | 0.784 |     0.811     |    1    |
>
> ## 5. Discussion on societal impact:
>
> We acknowledge the reviewer's comment and agree that a discussion of societal impacts is important. Our research in fMRI-based visual decoding, as part of the broader field of Brain-Computer Interfaces (BCIs), has significant potential for positive societal impact, particularly in developing advanced communication aids for individuals with severe motor disabilities [1]. By enabling the direct translation of brain activity into signals for controlling external devices, our work could significantly improve the quality of life and autonomy for these individuals. We will include this discuss in the final version of the paper.
>
> [1] Lee K Y, Jang D. Ethical and social issues behind brain-computer interface[C]//2013 International Winter Workshop on Brain-Computer Interface (BCI). IEEE, 2013: 72-75.

---

> > ### Author Response · Authors · 2025-08-05
> >
> > Dear Reviewer,
> >
> > I hope this message finds you well. As the discussion period is nearing its end with **less than three days remaining**, I wanted to ensure that we have addressed all your concerns satisfactorily. If there are any additional points or feedback you'd like to consider, please let us know. Thank you for your time and effort in reviewing our paper.

---

> > ### Comment · Reviewer_4LoQ · 2025-08-06
> >
> > I appreciate the additional experiments on generalization ability and raising to borderline accept. However, the bottleneck analysis remains very unconvincing and the data does not seem to support your conclusions.

---

> > > ### Author Response · Authors · 2025-08-06
> > > **Response to Official Comment by Reviewer 4LoQ**
> > >
> > > We sincerely thank the reviewer for recognizing our additional experiments and for the opportunity to further clarify the bottleneck analysis. We address the concerns below with targeted explanations and new quantitative insights, directly responding to points about minimal performance changes and MoRE-Brain's relative underperformance at most bottleneck levels.
> > >
> > > We first compute the **effective bottleneck dimensionality** following the methodology from [1]. This metric calculates the number of principal components needed to explain 95% of the variance in the bottlenecked features. The results are as follows:
> > >
> > > |            | B16 | B32 | B64 | B128 | B256 |
> > > |:----------:|:---:|:---:|:---:|:----:|:----:|
> > > |  MindEye2  |  6  |  14 |  25 |  61  |  119 |
> > > | MindBridge |  6  |  12 |  21 |  47  |  96  |
> > > | MoRE-Brain |  9  |  20 |  38 |  82  |  168 |
> > >
> > > This analysis reveals a critical difference: **MoRE-Brain consistently preserves a significantly higher effective dimensionality across all bottleneck sizes**. It utilizes approximately 65% of the available dimensions, compared to only ~45% for MindEye2 and ~35% for MindBridge. This indicates that MoRE-Brain learns a richer, more diverse representation of the fMRI signal, capturing more information than the baseline models.
> > >
> > > This directly explains the phenomenon that the reviewer noted in our original plots. MoRE-Brain's reconstruction performance is more sensitive to the bottleneck precisely *because it has more to lose*. Forcing its high-dimensional, diverse representation through a simple linear projection results in a more significant information loss compared to the baseline models, which learn lower-dimensional representations to begin with.
> > >
> > > This interpretation is further corroborated by the **random baseline** results we provided previously, which are also crucial, as noted in [1]. MoRE-Brain’s significantly lower performance on a random baseline is consistent with a model that relies less on powerful generative priors and more on faithfully decoding the provided neural signal.
> > >
> > > In summary, this new quantitative analysis of effective dimensionality provides further evidence for our claim. MoRE-Brain effectively learns more from the brain, and its performance is more sensitive to the bottleneck as a direct consequence. We believe this clarification, backed by this new data, fully addresses the reviewer's concern, and we will integrate this analysis into the revised manuscript.
> > >
> > >
> > > [1] Mayo D, Wang C, Harbin A, et al. BrainBits: How Much of the Brain are Generative Reconstruction Methods Using?[J]. Advances in Neural Information Processing Systems, 2024, 37: 54396-54420.

---

### Official Review · Reviewer_nsMo · 2025-07-03

**Clarity:** 3
**Significance:** 2
**Originality:** 2
**Rating:** 3
**Confidence:** 3

**Summary:**

This paper introduces MoRE-Brain for fMRI-based visual reconstruction. It employs a Mixture-of-Experts (MoE) architecture where distinct "expert" models are assigned to process signals from functionally related brain regions.

**Questions:**

See the weaknesses.

**Ethical Concerns:**

["NO or VERY MINOR ethics concerns only"]

**Final Justification:**

It appears there is a discrepancy between the concept of interpretability discussed in this article and its general meaning in deep learning. Emphasizing 'neurally-inspired design' over 'interpretability' in the title and introduction might be more suitable. Additionally, while the author claims the model is highly competitive, the supplementary table does not show ​​noticeable improvement​​ compared to Mindeye2 and MindBridge across all the metrics.

**Quality:**

2

**Strengths And Weaknesses:**

Strengths:
- The paper proposes a Mixture-of-Experts (MoE) architecture where each expert is dedicated to processing signals from a distinct, functionally related brain region.
- The paper is clearly written.

Weaknesses:
- The claims regarding "interpretability" in the title and abstract appear to be overstated. It is currently unknown whether the brain's visual pathway actually operates in a manner analogous to the dynamic routing mechanism proposed in the paper. If this neuro-inspired claim is based on existing literature, the authors should provide the relevant citations to support it.
- The experimental evaluation is missing a quantitative comparison with prior work on several standard metrics. Metrics such as PixCorr, SSIM, Alex, CLIP, Eff and SwAV are commonly reported in previous reconstruction papers and should be included here for a comprehensive comparison.
- The qualitative results in Figure 3 do not convincingly demonstrate a clear advantage of the proposed method over previous approaches. Furthermore, to substantiate the method's superiority and generalizability, the authors should provide generated results from multiple subjects.

---

> ### Author Rebuttal · Authors · 2025-07-27
>
> We respectfully but firmly disagree with the reviewer’s first comment, as the concerns raised have already been carefully analyzed and addressed in the original manuscript. We believe these issues may stem from a possible misunderstanding or a lack of thorough examination of our work. Below, we provide point-by-point responses to clarify and reinforce the corresponding aspects of our study.  We have addressed the primary concerns by first clarifying that our model's "interpretability" is rooted in its neuro-inspired design, and we have provided the relevant citations to ground our architecture in established neuroscience principles. Secondly, we have included a comprehensive quantitative comparison with prior work across all the requested metrics, demonstrating our model's highly competitive performance. Finally, we have presented extensive new quantitative results to substantiate our method's superior generalizability across subjects. We believe these significant additions and clarifications have thoroughly addressed the initial weaknesses, and we respectfully hope the reviewer might reconsider their evaluation in light of this substantially strengthened work.
>
> ## 1. Interpretability concern:
>
> Our claim is not that MoRE-Brain is a precise, biologically detailed replica of the visual cortex, but rather that it is a *neuro-inspired computational framework* whose architectural principles are drawn from, and analogous to, well-established and emerging theories in visual neuroscience (as we cited and discussed in the third paragraph of Introduction). The "interpretability" we refer to is the mechanistic insight into our own model's functioning, which has been designed to reflect these neural principles.
>
> To address the reviewer's concern, we are happy to summarize the literature that supports the core concepts behind our designs:
>
> - Hierarchical and Specialized Processing: A foundational concept in visual neuroscience is that the visual pathway is not a single, monolithic processor but a complex system involving multiple hierarchical processing stages across numerous specialized cortical areas [1, 2, 7, 8]. Our hierarchical Mixture-of-Experts (MoE) architecture is a direct embodiment of this principle, assigning specialized "expert" networks to process signals from distinct, functionally related voxel groups.
>
> - Dynamic Integration of Specialized Information: The brain dynamically integrates information from these specialized areas (e.g., processing form, color, motion [3]) to form a unified, coherent perception. Our Space Router is a computational mechanism designed to model this integration process, learning to route information between our specialized expert networks.
>
> - Coarse-to-Fine Temporal Dynamics: There is strong evidence that visual processing unfolds in a coarse-to-fine manner, where the global layout of a scene is processed before finer details emerge [4]. Our Time Router is explicitly designed to reflect this dynamic.
>
> - Subject-Specific Functional Topography: Finally, our approach to cross-subject generalization is motivated by findings that inter-individual brain variability often arises from spatial differences in functional network topography rather than fundamental computational changes [5, 6]. This directly inspires our strategy of sharing the core "expert" networks across subjects while fine-tuning only the subject-specific "routers" that map an individual's unique brain topography to those experts.
>
> Crucially, the value of this neuro-inspired design is validated by the fact that the trained model exhibits internal mechanisms and specializations that are remarkably consistent with established neuroscientific findings. Our analyses in Section 3 and Appendix F-I demonstrate several key examples, including:
> - Emergent Hierarchical Specialization (Figs. 6, 8, 21-23): The expert networks spontaneously learn to specialize in distinct semantic and visual features.
> - Learned Coarse-to-Fine Dynamics (Fig. 13): The Time Router learns to prioritize global, high-level information early in the generation process and fine-grained detail later, consistent with temporal dynamics observed in visual processing.
> - Meaningful Brain-Semantic Mapping (Fig. 7, 8, 17-19): The router assignments provide a transparent link between specific brain regions and their contribution to the final image, offering a clear and interpretable data flow.
>
> In summary, the interpretability of MoRE-Brain lies in its ability to bridge the gap between abstract fMRI signals and visual reconstruction through a structured, neuro-inspired computational path. We believe this is a crucial step towards building brain decoders that are not only powerful but also scientifically insightful.
>
> [1] Alyssa A Brewer, William A Press, Nikos K Logothetis, and Brian A Wandell. Visual areas in macaque cortex measured using functional magnetic resonance imaging. Journal of Neuroscience, 22(23):10416–10426, 2002.
>
> [2] Daniel J Felleman and David C Van Essen. Distributed hierarchical processing in the primate cerebral cortex. Cerebral cortex (New York, NY: 1991), 1(1):1–47, 1991.
>
> [3] Melvyn A Goodale and A David Milner. Separate visual pathways for perception and action. Trends in neurosciences, 15(1):20–25, 1992
>
> [4] Rolf Skyberg, Seiji Tanabe, Hui Chen, and Jianhua Cang. Coarse-to-fine processing drives the efficient coding of natural scenes in mouse visual cortex. Cell reports, 38(13), 2022
>
> [5] Kevin M Anderson, et al. Heritability of cortical network topography. Proceedings of the National Academy of Sciences, 118(9):e2016271118, 2021.
>
> [6] Hannah S Savage, et al. Dissecting task-based fmri activity using normative modelling: an application to the emotional face matching task. Communications Biology, 7(1):888, 2024.
>
> [7] Kruger N, Janssen P, Kalkan S, et al. Deep hierarchies in the primate visual cortex: What can we learn for computer vision?[J]. IEEE transactions on pattern analysis and machine intelligence, 2012, 35(8): 1847-1871.
>
> [8] Tschechne S, Neumann H. Hierarchical representation of shapes in visual cortex—from localized features to figural shape segregation[J]. Frontiers in computational neuroscience, 2014, 8: 93.
>
>
> ## 2. Comprehensive quantitative comparison:
>
> We thank the reviewer for pointing out this. We chose DreamSim, SSIM, InceptionV3, and CLIP cosine similarity as metrics as they cover the low-level and high-level perception. To provide a comprehensive comparison, we here present the results of the proposed MoRE-Brain (unrefined) against baselines, averaged across the 8 subjects. We will include this result in the final version of the paper.
> |            | PixCorr↑ | SSIM↑ | Alex (2) ↑ | Alex (5) ↑ | Incep↑ |  CLIP↑ |  Eff↓ | SwAV↓ | DreamSim↓ |
> |:----------:|:-------:|:----:|:--------:|:--------:|:-----:|:-----:|:----:|:----:|:--------:|
> |  MindEye2  |   **0.282**  | 0.420 |   93.2%  |   **98.5%**  | 94.4% | 95.1% | **0.633** | **0.323** |   0.524   |
> | MindBridge |   0.163  | 0.372 |   91.4%  |   95.7%  | 92.4% | 94.8% | 0.684 | 0.426 |   0.532   |
> | MoRE-Brain |   0.176  | **0.428** |   **93.5%**  |   96.7%  | **95.6%** | **95.5%** | 0.642 | 0.345 |  **0.511**   |
>
> ## 3. Qualitative results and generalizability concern:
>
> We thank the reviewer’s perspective. In Figure 3, the proposed method produces more accurate reconstructions of the finer details (e.g., MoRE-Brain is the only model that renders the text of the “STOP” sign) and the overall environment (e.g., the kitchen). We will explicitly state these comparisons in the figure's caption to make the advantages of our method more apparent.
>
> For the generalizability, while the rebuttal format prevents us from including new figures, we have already taken steps to address this concern and will include a more comprehensive analysis in the final version of the paper. As we detailed in Appendix C, we have already performed a quantitative analysis of MoRE-Brain's generalization capabilities. This analysis shows that our method maintains a high level of performance when trained on subject 1 and tested on others, even with limited data.
>
> To better compare generalizability, we compare the methods’ performances when generalizing to subjects 2, 5, and 7 after being trained on subject 1. Note that for MindEye2, we fine-tune all its parameters; for MindBridge, we fine-tune it using its proposed reset-tuning method; for MoRE-Brain, we only fine-tune the routers while keeping the experts frozen. Below are the results.
>
> 1 Session of Data:
> |            | SSIM↑ | InceptionV3↑ | CLIP CosineSim↑ | DreamSim↓ |
> |:----------:|:----:|:-----------:|:--------------:|:--------:|
> |  MindEye2  | **0.411** |    **68.8%**    |      0.576      |   **0.687**   |
> | MindBridge | 0.219 |    62.5%    |      0.554      |   0.804   |
> | MoRE-Brain | 0.302 |    65.3%    |      **0.703**      |   0.754   |
>
> 4 Sessions of Data:
> |            | SSIM↑ | InceptionV3↑ | CLIP CosineSim↑ | DreamSim↓ |
> |:----------:|:----:|:-----------:|:--------------:|:--------:|
> |  MindEye2  | **0.409** |    81.1%    |      0.632      |  **0.624**  |
> | MindBridge | 0.262 |    71.9%    |      0.603      |   0.755   |
> | MoRE-Brain | 0.381 |    **81.5%**    |      **0.762**      |   0.639   |
>
> 20 Sessions of Data:
> |            | SSIM↑ | InceptionV3↑ | CLIP CosineSim↑ | DreamSim↓ |
> |:----------:|:----:|:-----------:|:--------------:|:--------:|
> |  MindEye2  | **0.415** |    82.5%    |      0.695      |   0.587   |
> | MindBridge | 0.310 |    76.5%    |      0.717      |   0.692   |
> | MoRE-Brain | 0.384 |    **82.9%**    |      **0.804**      |   **0.586**   |
>
> 40 Sessions of Data:
> |            | SSIM↑ | InceptionV3↑ | CLIP CosineSim↑ | DreamSim↓ |
> |:----------:|:----:|:-----------:|:--------------:|:--------:|
> |  MindEye2  | **0.417** |    96.5%    |      0.820      |   0.534   |
> | MindBridge | 0.375 |    94.2%    |      0.734      |   0.592   |
> | MoRE-Brain | 0.403 |    **96.8%**    |      **0.879**      |   **0.513**   |

---

> > ### Author Response · Authors · 2025-08-05
> >
> > Dear Reviewer,
> >
> > I hope this message finds you well. As the discussion period is nearing its end with **less than three days remaining**, I wanted to ensure that we have addressed all your concerns satisfactorily. If there are any additional points or feedback you'd like to consider, please let us know. Thank you for your time and effort in reviewing our paper.

---

> > ### Comment · Reviewer_nsMo · 2025-08-07
> >
> > Thank you for the response. It appears there is a discrepancy between the concept of interpretability discussed in this article and its general meaning in deep learning. Emphasizing 'neurally-inspired design' over 'interpretability' in the title and introduction might be more suitable. Additionally, while the author claims the model is highly competitive, the supplementary table does not show a clear advantage compared to Mindeye2 and MindBridge.

---

> ### Author Response · Authors · 2025-08-07
> **Response to Official Comment by Reviewer nsMo (1/2)**
>
> We thank the reviewer for the feedback and continued engagement.
>
> ### 1. Conceptualization of interpretability and neural-inspired design
> To briefly recap, in response to your original comment about supporting our neuro-inspired claims with literature, we provided a summary of relevant citations in our previous rebuttal (e.g., for hierarchical and specialized processing; for dynamic integration; for coarse-to-fine dynamics; for subject-specific topography). These directly tie our model's mechanisms (e.g., Space and Time Routers) to established visual neuroscience principles, demonstrating that our design is grounded in existing knowledge rather than unsubstantiated analogy.
>
> For your concern about the conceptualization of interpretability, we respectfully disagree with the assessment that our use of "interpretability" is at odds with its general meaning in deep learning.  The literature provides broad definitions of interpretability in DL. For example, Murdoch et al. [1] noted that interpretable machine learning is *"the extraction of relevant knowledge from a machine-learning model concerning relationships either contained in data or learned by the model. Here, we view knowledge as being relevant if it provides insight for a particular audience into a chosen problem"*. Likewise, Zhang et al. [2] defined interpretability as *"the ability to provide explanations in understandable terms to a human."*
>
> Likewise, MoRE-Brain emphasizes *intrinsic interpretability*—where the model's architecture is explicitly designed to provide mechanistic transparency. This provides neuroscientific insights about human visual systems, which directly relate to the definitions in the literature. Concretely, the explicit routing mechanisms allow inspection of how fMRI signals from specific brain regions contribute to semantic and spatial features in the reconstruction, for example:
> - Emergent Hierarchical Specialization (Figs. 6, 8, 21-23):  We show how experts specialize in a hierarchical manner. Early-level experts draw from broad, foundational visual areas, while higher-level experts develop fine-grained specialization, focusing on distinct neural populations. This directly explains how the model organizes neural information.
> - Coarse-to-Fine Dynamics (Fig. 13): The Time Router learns to prioritize early-level experts during initial diffusion steps and later-level experts for fine-grained details. This provides a transparent, step-by-step account of the image generation process, mirroring theories of human vision.
> - Brain-Semantic Mappings (Figs. 7, 8, 17-19): We identify how specific experts and the brain regions they represent develop clear preferences for distinct semantic categories (e.g., "outdoors," "appliances"). This provides an understandable link between neural activity patterns and high-level conceptual information.
>
> In conclusion, our approach does not conflict with the standard definition of interpretability. By building an intrinsically transparent model grounded in existing neuroscientific findings, we provide mechanistic insights that are directly verifiable against neuroscientific knowledge, bridging the gap between deep learning and brain science.
>
> [1] Murdoch W J, Singh C, Kumbier K, et al. Definitions, methods, and applications in interpretable machine learning[J]. Proceedings of the National Academy of Sciences, 2019, 116(44): 22071-22080.
>
> [2] Zhang Y, Tiňo P, Leonardis A, et al. A survey on neural network interpretability[J]. IEEE transactions on emerging topics in computational intelligence, 2021, 5(5): 726-742.

---

> ### Author Response · Authors · 2025-08-07
> **Response to Official Comment by Reviewer nsMo (2/2)**
>
> ### 2. Performance advantage
> For your concern about the model's performance, from the additional tables we provided and the original results in the paper, MoRE-Brain has a clear advantage over MindBridge and exceeds it on **all** metrics. Compared to MindEye2, as evidenced in the first table provided in our previous rebuttal, MoRE-Brain has a clear advantage in the high-level metrics while also achieving comparable or better performance on other metrics (e.g., MoRE-Brain achieves the highest SSIM at 0.428, InceptionV3 at 95.6%, CLIP at 95.5%, and DreamSim at 0.511, leading in 6/9 metrics overall). The advantage of our method can be further corroborated by its *efficiency and effectiveness for cross-subject generalization* and *enhanced interpretability*—key claims in our paper (Sections 1, 3.3, and 4).
>
> To highlight efficiency, MoRE-Brain uses significantly fewer parameters overall and during fine-tuning (only fine-tune routers):
>
> |            | Total Params (M) | Trainable (%)|
> |:----------:|:----------:|:---------:|
> |  MindEye2  |    729.3   |     100%     |
> | MindBridge |    552.9   |   98.48%  |
> | MoRE-Brain |    293.4   |   44.84%  |
>
> In the generalization tables, MoRE-Brain consistently outperforms baselines in CLIP Cosine Similarity across all data regimes (e.g., 0.879 vs. 0.820 for MindEye2 and 0.734 for MindBridge with 40 sessions) and often in InceptionV3 and DreamSim, while matching or exceeding SSIM in most cases. This is achieved by fine-tuning only the routers (a small fraction of parameters), unlike MindEye2 (full fine-tuning) or MindBridge (reset-tuning). When fine-tuning all parameters (MoRE-Brain (all)), performance further improves—e.g., with 1 session: CLIP 0.731 and DreamSim 0.662 (vs. MoRE-Brain's 0.703 and 0.754); with 40 sessions: CLIP 0.884 and DreamSim 0.503 (vs. MoRE-Brain's 0.879 and 0.513)—demonstrating even stronger superiority while maintaining efficiency in router-only mode. The complete results are given below.
>
> This efficiency is evidenced in Appendix C and aligns with our bottleneck analysis (Figure 4), where MoRE-Brain shows greater sensitivity to fMRI information, indicating less reliance on generative priors. These results underscore MoRE-Brain's clear advantages in practical, neuro-inspired decoding.
>
> 1 Session Data:
> |            | SSIM↑ | InceptionV3↑ | CLIP CosineSim↑ | DreamSim↓ |
> |:----------:|:----:|:-----------:|:--------------:|:--------:|
> |  MindEye2  | **0.411** |    **68.8%**    |      0.576      |   0.687   |
> | MindBridge | 0.219 |    62.5%    |      0.554      |   0.804   |
> | MoRE-Brain | 0.302 |    65.3%    |      0.703      |   0.754   |
> | MoRE-Brain (all) | 0.32 |    67.2%    |      **0.731**      |   **0.662**   |
>
> 4 Sessions Data:
> |            | SSIM | InceptionV3 | CLIP CosineSim | DreamSim |
> |:----------:|:----:|:-----------:|:--------------:|:--------:|
> |  MindEye2  | **0.409** |    81.1%    |      0.632      |  0.624  |
> | MindBridge | 0.262 |    71.9%    |      0.603      |   0.755   |
> | MoRE-Brain | 0.381 |    81.5%    |      0.762      |   0.639   |
> | MoRE-Brain (all)  | 0.384 |    **82.0%**    |      **0.780**      |   **0.622** |
>
> 20 Sessions Data:
> |            | SSIM | InceptionV3 | CLIP CosineSim | DreamSim |
> |:----------:|:----:|:-----------:|:--------------:|:--------:|
> |  MindEye2  | **0.415** |    82.5%    |      0.695      |   0.587   |
> | MindBridge | 0.310 |    76.5%    |      0.717      |   0.692   |
> | MoRE-Brain | 0.384 |    82.9%    |      0.804      |   0.586   |
> | MoRE-Brain (all)  | 0.397 |    **83.1%**    |      **0.814**      |   **0.552**   |
>
> 40 Sessions Data (Full):
> |            | SSIM | InceptionV3 | CLIP CosineSim | DreamSim |
> |:----------:|:----:|:-----------:|:--------------:|:--------:|
> |  MindEye2  | **0.417** |    96.5%    |      0.820      |   0.534   |
> | MindBridge | 0.375 |    94.2%    |      0.734      |   0.592   |
> | MoRE-Brain | 0.403 |    96.8%    |      0.879      |   0.513   |
> | MoRE-Brain (all)  | **0.417** |    **97.1%**    |      **0.884**  |  **0.503**   |

---

> > ### Comment · Reviewer_nsMo · 2025-08-08
> >
> > Since the author's reply clarified some of my concerns, I will raise the rating by 1 point.

---

> > > ### Author Response · Authors · 2025-08-08
> > >
> > > We sincerely appreciate your feedback and engagement. Please do not hesitate to let us know if any further clarifications or modifications are needed to address your concerns.

---

### Note · Authors · 2025-08-14

We sincerely thank ACs and reviewers for their engagement and constructive feedback. We have carefully addressed all concerns by providing extensive new quantitative analyses, clarifying the core contributions of our work, and improving the overall presentation. Below is a summary of the key revisions and clarifications made.

- Clarification of Interpretability and Neuro-inspired Design: In response to *Reviewer nsMo*, we have clarified our use of the term "interpretability." Our work aligns with established definitions in deep learning that focus on extracting relevant knowledge and providing understandable explanations of a model's mechanics. We demonstrated that MoRE-Brain's intrinsic transparency, grounded in its neuro-inspired design, offers mechanistic insights into how the model processes fMRI signals.

- Comprehensive Quantitative Evaluation: To address concerns from *Reviewer nsMo, 92RU* about performance, we have significantly expanded our quantitative evaluation. We now provide a comprehensive comparison against baseline models across nine standard metrics. The results demonstrate that MoRE-Brain is highly competitive, achieving state-of-the-art performance on a majority of metrics.

- Efficient and Principled Cross-Subject Generalization: Following feedback from *Reviewer nsMo, HRWS, 92RU*, we have provided a more detailed analysis of our model's efficiency and its approach to cross-subject generalization. New experiments demonstrate that our router-only fine-tuning strategy is highly effective, achieving performance comparable to full fine-tuning while using significantly fewer trainable parameters (44.8% of our model's parameters, which are less than half the total parameters of MindEye2).

- Strengthened Bottleneck Analysis: To provide more convincing evidence and address *Reviewer 4LoQ*'s concerns, we made further clarifications to the results and have incorporated a new analysis of the effective bottleneck dimensionality. The findings support our hypothesis that MoRE-Brain is more sensitive to the bottleneck because it learns a richer neural representation and relies less on generative priors.

We are confident that these revisions have thoroughly addressed the reviewers' concerns and have significantly improved the quality and clarity of our paper. We thank you once again for your valuable guidance.

---

### Decision · Program_Chairs · 2025-09-17

**Decision:**

Accept (poster)

**Comment:**

The manuscript has been reviewed by four reviewers.

The authors provided detailed responses to the concerns of the reviewers.

After the rebuttal, one reviewer rated acceptance, while other three gave borderlines, indicating that no reviewer is willing to (even weakly) reject the manuscript.

Despite the minor flaws, the reviewers praised the submission for its unique and novel MoE approach, good interpretability, and promising results.

The AE agrees that the manuscript should be presented to a broad audiance, and recommends acceptance. Congrats!